# QUICKEST CHANGE DETECTION FOR MULTI-TASK PROBLEMS UNDER UNKNOWN PARAMETERS

## ABSTRACT

We consider the quickest change detection problem where both the parameters of pre- and post- change distributions are unknown, which prevent the use of classical simple hypothesis testing. Without additional assumptions, optimal solutions are not tractable as they rely on some minimax and robust variant of the objective. As a consequence, change points might be detected too late for practical applications (in economics, health care or maintenance for instance). Other approaches solve a relaxed version of the problem through the use of particular probability distributions or the use of domain knowledge. We tackle this problem in the more complex Markovian case and we provide a new scalable approximate algorithm with near optimal performance that runs in $\mathcal{O}(1)$.

## 1 INTRODUCTION

Quickest Change Detection (QCD) problems arise naturally in settings where a latent state controls observable signals (Basseville et al., 1993). In biology, it is applied in genomic sequencing (Caron et al., 2012) and in reliable healthcare monitoring (Salem et al., 2014). In industry, it finds application in faulty machinery detection (Lu et al., 2017; Martí et al., 2015) and in leak surveillance (Wang et al., 2014). It also has environmental applications such as traffic-related pollutants detection (Carslaw et al., 2006).

Any autonomous agent designed to interact with the world and achieve multiple goals must be able to detect relevant changes in the signals it is sensing in order to adapt its behaviour accordingly. This is particularly true for reinforcement learning based agents in multi-task settings as their policy is conditioned to some task parameter (Gupta et al., 2018; Teh et al., 2017). In order for the agent to be truly autonomous, it needs to identify the task at hand according to the environment requirement. For example, a robot built to assist cooks in the kitchen should be able to recognise the task being executed (chopping vegetables, cutting meat, ..) without external help to assist them efficiently. Otherwise, the agent requires a higher intelligence (one of the cooks for instance) to control it (by stating the task to be executed). In the general case, the current task is unknown and has to be identified sequentially from external sensory signals. The agent must track the changes as quickly as possible to adapt to its environment. However, current solutions for the QCD problem when task parameters are unknown, either do not scale or impose restrictive conditions on the setting. (i.i.d. observations, exponential family distributions, partial knowledge of the parameters, etc.).

In this paper, we construct a scalable algorithm with similar performances to optimal solutions. For this purpose, we use the change detection delay under known parameters as a lower bound for the delay in the unknown case. This improves our estimations of the parameters and thus improves our change point detection. We consider the case where the data is generated by some Markovian processes as in reinforcement learning. We assess our algorithm performances on synthetic data generated using distributions parameterised with neural networks in order to match the complexity level of real life applications. We also evaluate our algorithm on standard reinforcement learning environment.

## 2 QUICKEST CHANGE DETECTION PROBLEMS

Formally, consider a sequence of random observations $(X_t)$ where each $X_t$ belongs to some observation space $\mathcal{X}$ (say, an Euclidean space for simplicity) and is drawn from $f_{\theta_t}(.|X_{t-1})$, where

the parameter $\theta_t$ belongs to some task parameter space $\Theta$ and $\{f_\theta, \theta \in \Theta\}$ is a parametric probability distribution family (non trivial, in the sense that all $f_\theta$ are different). The main idea is that, at almost all stages, $\theta_{t+1} = \theta_t$ but there are some "change points" where those two parameters differ. Let us denote by $t_k$ the different change points and, with a slight abuse of notations, by $\theta_k$ the different values of the parameters. Formally, the data generating process is therefore: $X_t \sim \sum_{k=0}^K f_{\theta_k}(.|X_{t-1}) \mathbb{1}_{t_k \le . < t_{k+1}}(t)$.

The overarching objective is to identify as quickly as possible the change points $t_k$ and the associated parameters $\theta_k$, based on the observations $(X_t)$. Typical procedures propose to tackle iteratively the simple change point detection problem, and for this reason we will focus mainly on the simpler setting of a single change point, where $K = 2$, $t_0 = 0$, $t_1 = \lambda$ and $t_2 = \infty$, where $\lambda$ is unknown and must be estimated. For a formal description of the model and the different metrics, we will also assume that the parameters $(\theta_0, \theta_1)$ are drawn from some distribution $\mathcal{F}$ over $\Theta$. As a consequence, the data generating process we consider boils down to the following system 1:

$$\theta_0, \theta_1 \sim \mathcal{F} \quad \text{and} \quad \begin{cases} X_{t+1} & \sim & f_{\theta_0}(.|X_t) & \text{if} & t \le \lambda \\ X_{t+1} & \sim & f_{\theta_1}(.|X_t) & \text{if} & t > \lambda \end{cases}. \tag{1}$$

## 2.1 CRITERIA DEFINITIONS

As mentioned, the objective is to detect change points as quickly as possible while controlling the errors. There exist different metrics to evaluate algorithm performances; they basically all minimise some delay measurements while keeping the rate of type I errors (false positive change detection) under a certain level. We will describe later on different existing definitions of these criteria (type I error and delay). In order to evaluate a probability of error and an expected delay, we obviously need to define relevant probability measures first. Traditionally, there are two antagonistic ways to construct them: the MIN-MAX and the BAYESIAN settings (Veeravalli & Banerjee, 2014).

First, we denote by $\mathbb{P}_n$ (resp. $\mathbb{E}_n$) the data probability distribution (resp. the expectation) conditioned to the change point happening at $\lambda = n$. This last event happens with some probability $\mu(\boldsymbol{\lambda} = n)$ – with the notation that bold characters designate random variables–, and we denote by $\mathbb{P}^\mu$ (resp. $\mathbb{E}^\mu$) the data probability distribution (resp. the expectation), integrated over $\mu$, *i.e.*, for any event $\Omega$, it holds that $\mathbb{P}^\mu(\Omega) = \sum_n \mu(\boldsymbol{\lambda} = n) \mathbb{P}_n(\Omega)$. In the following, we describe the major existing formulations of the QCD problem where the goal is to identify an optimal stopping time $\boldsymbol{\tau}$:

**BAYESIAN formulation:** In this formulation, the error is the Probability of False Alarms (**PFA**) with respect to $\mathbb{P}^\mu$. The delay is evaluated as the Average Detection Delay (**ADD**) with respect to $\mathbb{E}^\mu$.

$$\mathbf{PFA}(\boldsymbol{\tau}) = \mathbb{P}^\mu(\boldsymbol{\tau} < \lambda) \quad \text{and} \quad \mathbf{ADD}(\boldsymbol{\tau}) = \mathbb{E}^\mu[(\boldsymbol{\tau} - \lambda)|\boldsymbol{\tau} > \lambda]$$

In this setting, the goal is to minimise **ADD** while keeping **PFA** below a certain level $\alpha$ (as in Shiryaev formulation (Shiryaev, 1963)). Formally, this rewrites into:

$$(\text{SHIRYAEV}) \begin{cases} \boldsymbol{\nu}_\alpha = \arg\min_{\boldsymbol{\tau} \in \Delta_\alpha} \mathbf{ADD}(\boldsymbol{\tau}) \\ \Delta_\alpha = \{\boldsymbol{\tau} : \mathbf{PFA}(\boldsymbol{\tau}) < \alpha\} \end{cases} \tag{2}$$

**Min-Max formulation:** The MIN-MAX formulation disregards prior distribution over the change point. As a consequence, the error is measured as the False Alarm Rate (**FAR**) with respect to the worst case scenario where no change occurs ($\mathbb{P}_\infty$). As for the delay, two possibilities are studied: the Worst Average Detection Delay (**WADD**) and the Conditional Average Detection Delay (**CADD**). **WADD** evaluates this delay with respect to the worst scenario in terms of both the change point and the observations. **CADD** is a less pessimistic evaluation as it only considers the worst scenario in terms of change point. Mathematically they are defined as:

$$\mathbf{FAR}(\boldsymbol{\tau}) = \frac{1}{\mathbb{E}_\infty[\boldsymbol{\tau}]} \quad \text{and} \quad \begin{cases} \mathbf{WADD}(\boldsymbol{\tau}) = \sup_n \operatorname{esssup}_{X^n} \mathbb{E}_n[(\boldsymbol{\tau} - n)^+ | X^n] \\ \mathbf{CADD}(\boldsymbol{\tau}) = \sup_n \mathbb{E}_n[(\boldsymbol{\tau} - n)|\boldsymbol{\tau} > \lambda] \end{cases}$$

where $X^n$ designates all observation up to the $n^{th}$ one. In this setting, the goal is to minimise either **WADD** (Lorden formulation (Lorden et al., 1971)); or **CADD** (Pollak formulation (Pollak, 1985)) while keeping **FAR** below a certain level $\alpha$. Formally, these problems are written as follows:

$$(\text{LORDEN}) \begin{cases} \boldsymbol{\nu}_\alpha = \arg\min_{\boldsymbol{\tau} \in \Delta_\alpha} \mathbf{WADD}(\boldsymbol{\tau}) \\ \Delta_\alpha = \{\boldsymbol{\tau} : \mathbf{FAR}(\boldsymbol{\tau}) < \alpha\} \end{cases} \quad \text{and} \quad (\text{POLLAK}) \begin{cases} \boldsymbol{\nu}_\alpha = \arg\min_{\boldsymbol{\tau} \in \Delta_\alpha} \mathbf{CADD}(\boldsymbol{\tau}) \\ \Delta_\alpha = \{\boldsymbol{\tau} : \mathbf{FAR}(\boldsymbol{\tau}) < \alpha\} \end{cases} \tag{3}$$

## 3 TEMPORAL WEIGHT REDISTRIBUTION

Under known parameters, optimal solutions for the QCD consist in computing some statistics, denoted by $S_n^{\theta_0,\theta_1}$ along with a threshold $B_\alpha$. The stopping time $\nu_\alpha$ is the first time $S_n^{\theta_0,\theta_1}$ exceeds $B_\alpha$. We provide a more detailed description in Appendix A.1.

### 3.1 ASYMPTOTIC BEHAVIOUR OF THE SOLUTION TO THE BAYESIAN FORMULATION

The SHIRYAEV algorithm (Shiryaev, 1963) is asymptotically optimal in the i.i.d. case (Tartakovsky & Veeravalli, 2005). This result is extended to the non i.i.d. case under the following assumption:

**Hypothesis 1** *Given $f_{\theta_0}$ and $f_{\theta_1}$, there exists $q \in \mathbb{R}$ and $r \in \mathbb{N}$ such that for any $k \in \mathbb{N}$:*

$$\frac{1}{n}\sum_{t=k}^{k+n} \frac{f_{\theta_1}(X_{t+1}|X_t)}{f_{\theta_0}(X_{t+1}|X_t)} \xrightarrow[n\to+\infty]{r\text{-}quickly} q \tag{4}$$

Under Hypothesis 1, and with exponential or heavy tail prior, the SHIRYAEV algorithm is asymptotically optimal (Tartakovsky & Veeravalli, 2005). In addition, the moments of **ADD** satisfy the following property for all $m \le r$ (Tartakovsky & Veeravalli, 2005):

$$\mathbb{E}^\mu[(\boldsymbol{\nu}_\alpha - \lambda)^m | \tau > \lambda] \overset{\alpha\to 0}{\sim} \mathbb{E}^\mu[(\boldsymbol{\nu}_\alpha^S - \lambda)^m] \overset{\alpha\to 0}{\sim} \left(\frac{|\log(\alpha)|}{q+d}\right)^m \tag{5}$$

where $\boldsymbol{\nu}_\alpha^S$ is the SHIRYAEV algorithm stopping rule and $d = -\lim_{n\to\infty} \log \mathbb{P}(\lambda > n+1)/n$. Our approach to the QCD problem under unknown parameters relies on the asymptotic behaviour from Equation (5) up to the second order ($r = 2$). We use it as a lower bound for the detection delay, providing a natural segmentation to our data when approximating the parameters. We will discuss this more in details in what follows.

### 3.2 RATIONAL FOR TEMPORAL WEIGHT REDISTRIBUTION

In the following, we denote the true parameters by $\theta_0^*$ and $\theta_1^*$ and the corresponding stationary distributions by $\Pi_0^*$ and $\Pi_1^*$ (that are well defined for irreducible Markov Chain). The purpose of this section is to devise an algorithm learning the parameters $\theta_0^*$ and $\theta_1^*$ using the asymptotic behaviour of the detection delay under known parameters. The delay predicted in Equation (5) is a performance lower bound in our setting (otherwise it would contradict the optimality of the SHIRYAEV algorithm). Intuitively, the idea is to learn $\theta_1^*$ using the last $\frac{|\log(\alpha)|}{q+d}$ observation and to learn $\theta_0^*$ using previous observation. This simple technique happens to be efficient and computationally simple.

**Parameters optimisation:** The optimal $S_t^{\theta_0,\theta_1}$ – called the SHIRYAEV statistic – has a specific form (see Equation (12) in Appendix) and the QCD problem under unknown parameters is equivalent to:

$$\nu_\alpha = \arg\min_t\{t | S_t^{\theta_0,\theta_1} > B_\alpha\} \quad \text{where} \quad \begin{cases} \theta_0 = \arg\min_\theta f_0(\theta, \theta^1) \\ \theta_1 = \arg\max_\theta f_1(\theta^0, \theta) \end{cases}, \tag{6}$$

and $(\theta^0, \theta^1)$ are random initialisation parameters, $f_0(\theta_0, \theta_1) = \mathbb{E}_0\left[\log \frac{f_{\theta_1}(X_{t+1}|X_t)}{f_{\theta_0}(X_{t+1}|X_t)}\right]$, $f_1(\theta_0, \theta_1) = \mathbb{E}_1\left[\log \frac{f_{\theta_1}(X_{t+1}|X_t)}{f_{\theta_0}(X_{t+1}|X_t)}\right]$, and $\mathbb{E}_{k\in\{0,1\}}$ is the expectation with respect to the probability distribution $\mathbb{P}_k(X_t, X_{t+1}) = \Pi_k^*(X_t)f_{\theta_k^*}(X_{t+1}|X_t)$. In fact, the solution $\nu_\alpha$ to Equation (6) is the optimal stopping time under the true parameters. This is a consequence of the following Lemma 1:

**Lemma 1** $\theta_0^*$ and $\theta_1^*$ *verifies the following for any $\theta_0, \theta_1$:*

$$\theta_0^* = \arg\min_\theta f_0(\theta, \theta_1) \quad and \quad \theta_1^* = \arg\max_\theta f_1(\theta_0, \theta) \tag{7}$$

In order to solve the equivalent QCD problem of Equation (6), a good approximation of the functions $f_0$ and $f_1$ given the observations $(X_t)_0^n$ is required. This implies the ability to distinguish pre-change samples from post-change samples and this is precisely why optimal solutions are intractable. They compute the statistics for any possible change point and consider the worst case scenario. Previous approximate solutions simplify the setting by using domain knowledge to infer the pre-change

distribution and by using a sliding window to evaluate the post-change distribution. The window size $w$ is an irreducible cost in the expected delay.

With known parameters, optimal algorithms have an average detection delay proportional to the error threshold $\alpha$ and to the inverse of the KL divergence between the pre and post change distributions. Given the optimal stopping time $\nu_\alpha$, it's possible to evaluate the posterior distribution of the change point $\mathbb{P}(\boldsymbol{\lambda} = t|\nu_\alpha = n)$, which in turn is a good classifier of the pre and post change observation:

$$\mathbb{P}(X_t \sim f_{\theta_0}|\nu_\alpha = n) \propto \mathbb{P}(\boldsymbol{\lambda} > t|\nu_\alpha = n) \; ; \; \mathbb{P}(X_t \sim f_{\theta_1}|\nu_\alpha = n) \propto \mathbb{P}(\boldsymbol{\lambda} < t|\nu_\alpha = n) \; . \quad (8)$$

The objective of this section is to exploit this classifier to construct a family of functions that approximate well $f_0$ and $f_1$. Consider the following family of functions:

$$f_0^n(\theta_0, \theta_1) = \mathbb{E}_0^n\big[\log(\tfrac{f_{\theta_1}}{f_{\theta_0}})(X_{\tau+1}|X_\tau)\big] \text{ and } f_1^n(\theta_0, \theta_1) = \mathbb{E}_1^n\big[\log(\tfrac{f_{\theta_1}}{f_{\theta_0}})(X_{\tau+1}|X_\tau)\big] \; ,$$

where both the observations $X_t$ and the indicies $\tau$ are random variables. The observations are generated using Equation (1) with a finite horizon $t_2$. The indicies are sampled with respect to $\tau \sim \mathbb{P}(\boldsymbol{\lambda} > \tau|\nu_\alpha = n)/\sum_{i=0}^{t_2} \mathbb{P}(\boldsymbol{\lambda} > i|\nu_\alpha = n)$ in $\mathbb{E}_0^n$ while they are sampled with respect to $\tau \sim \mathbb{P}(\boldsymbol{\lambda} < \tau|\nu_\alpha = n)/\sum_{i=0}^{t_2} \mathbb{P}(\boldsymbol{\lambda} < i|\nu_\alpha = n)$ in $\mathbb{E}_1^n$.

The family of functions $f_k^n$ are a re-weighted expectation of the log-likelihood ratio of the observations using $\mathbb{P}(\boldsymbol{\lambda}|\nu_\alpha = n)$. Basically, under the assumption that the optimal detection happens at $t = n$, we associate to each observation $X_\tau$ a weight proportional to $\mathbb{P}(\boldsymbol{\lambda} > \tau|\nu_\alpha = n)$ when estimating $f_0$ (and respectively proportional to $\mathbb{P}(\boldsymbol{\lambda} < \tau|\nu_\alpha = n)$ when estimating $f_1$). In addition, this family of functions is practical as we can approximate asymptotically $\mathbb{P}(\boldsymbol{\lambda}|\nu_\alpha = n)$ using the theoretical delay behaviour of the SHIRYAEV algorithms presented in Equation (5).

**Lemma 2** *For any given parameters $(\theta_0, \theta_1)$, the following convergences hold:*

*If $\lambda = \infty$, then:* $\lim_{n,t_2\to\infty} |f_0^n(\theta_0, \theta_1) - f_0(\theta_0, \theta_1)| = \lim_{n,t_2\to\infty} |f_1^n(\theta_0, \theta_1) - f_0(\theta_0, \theta_1)| = 0$

*If $\lambda < \infty$, then:*

$$\begin{cases} \lim_{n,t_2\to\infty} |f_0^n(\theta_0, \theta_1) - f_1(\theta_0, \theta_1)| & = & 0 \\ \lim_{n,t_2\to\infty} |f_1^n(\theta_0, \theta_1) - f_1(\theta_0, \theta_1)| & = & 0 \\ \lim_{t_2\to\infty} |f_1^{\nu_\alpha}(\theta_0, \theta_1) - f_1(\theta_0, \theta_1)| & = & 0 \end{cases}$$

*For any integer $n \in \mathbb{N}$, if $t_2 = \lambda + n$, then:* $\lim_{\lambda,t_2\to\infty} |f_0^{\nu_\alpha} - f_0| = 0$.

A major implication of Lemma 2 is that with enough observations, $f_0^{\nu_\alpha}$ and $f_1^{\nu_\alpha}$ will eventually converge to the functions $f_0$ and $f_1$. In fact, $f_0^t$ approaches $f_0$ up to $\nu_\alpha$ and then start degrading (as it ends up converging to $f_1$) whereas $f_1^t$ becomes a better and better approximation of $f_1$ around $\nu_\alpha$ and improves asymptotically. As such, we are going to consider the following problem as proxy to the original QCD problem (Equation (6)).

$$\hat{\nu}_\alpha = \arg\min_t \{t|S_t^{\theta_0^t, \theta_1^t} > B_\alpha\} \quad \text{where} \quad \begin{cases} \theta_0^t = \arg\min_\theta f_0^t(\theta, \theta_1^{t-1}) \\ \theta_1^t = \arg\max_\theta f_1^t(\theta_0^{t-1}, \theta) \end{cases} . \quad (9)$$

The rational behind this choice is that around the change point, $S_t^{\theta_0^t, \theta_1^t}$ converges to $S_t^{\theta_0^*, \theta_1^*}$. Having access to a perfect evaluation of the functions $f_0$ and $f_1$ guarantees the fastest possible detection. In fact, $\nu_\alpha$ – the optimal stopping time under known parameters – is a lower bound to the detection delay in this setting. In practice, only the first $t$ observation are accessible to evaluate $f_0^t$ and $f_1^t$. This introduces approximation errors to the estimates $\theta_0^t$ and $\theta_1^t$, thus delaying the detection. However, this is not detrimental to the evaluation of the stopping time. Indeed:

**Before $\nu_\alpha$:** $\theta_0^t$ is a good estimator of $\theta_0^*$ and $\theta_1^t$ is a bad estimator of $\theta_1^*$. In fact, the fraction of pre-change observations used to learn $\theta_1^*$ is of the order of $\min(1, \lambda/t)$. This helps maintaining a low log-likelihood ratio (equivalently maintain $S_t^{\theta_0^t, \theta_1^t}$ below the cutting threshold), as the estimation of $\theta_1^*$ will converge to a parameter close to $\theta_0^*$.

**After $\nu_\alpha$:** $\theta_1^t$ is a good approximation of $\theta_1^*$, but $\theta_0^t$ is a noisy estimation of $\theta_0^*$ (as post-change observations are used to learn it). The fraction of the data generated using $\theta_1^*$ but used to learn $\theta_0^*$ is proportional to $\frac{t-\nu_\alpha}{t}$. This favours – up to a certain horizon – higher log-likelihood ratio estimates (equivalently an incremental behaviour of the sequence $S_t^{\theta_0^t, \theta_1^t}$). We will discuss further the performance issues of solving Equation (9) in the following.

**Distribution Approximation:** In order to exploit the previously introduced results, we need a good approximation of $\mathbb{P}(\boldsymbol{\lambda}|\nu_\alpha = n)$. If the observation satisfy Hypothesis 1, then the moments of $\mathbb{P}(\boldsymbol{\lambda}|\nu_\alpha = n)$ up to the $r^{\text{th}}$ order satisfy Equation (5).

When $r = \infty$, this is equivalent to the Hausdorff moment problem, which admits a unique solution. The generalised method of moments can be used to approximate this distribution. In the remaining of this paper, we will restrict ourselves to the case where $r = 2$. There is an infinite number of distributions that satisfy Equation (5) in this case; however, given a parametric family, we can compute analytically a solution of the problem so we consider the Logistic distribution. For a given $\theta_0, \theta_1$, we approximate $\mathbb{P}(\boldsymbol{\lambda}|\nu_\alpha = n)$ with $f_n^{\theta_0,\theta_1} = \text{Logistic}(\mu, s)$ where $\mu = n - \frac{|\log(\alpha)|}{D_{KL}(f_{\theta_0}|f_{\theta_1})+d}$ and $s = \sqrt{3}\frac{|\log(\alpha)|}{\pi(D_{KL}(f_{\theta_0}|f_{\theta_1})+d)}$. Verifying that $f_n^{\theta_0,\theta_1}$ satisfies Equation (5) for $m \leq 2$ is straightforward. As a consequence, $\mathbb{P}(\boldsymbol{\tau} > t|\boldsymbol{\tau} \sim f_n^{\theta_0,\theta_1})$ is a fair approximation of $\mathbb{P}(\boldsymbol{\lambda} > t|\nu_\alpha = n)$.

**Limitations:** Solving the problem given by Equation (9) unfortunately yields poor performances for extreme values of the error threshold $\alpha$. This is due to a degraded log likelihood estimation both before $\lambda$ and as $t \to \infty$. In fact, the log-likelihood ratios $L_t$ and $L_t^*$ defined as:

$$L_t(X_{t+1}|X_t) = \log \frac{f_{\theta_1^t}}{f_{\theta_0^t}}(X_{t+1}|X_t) \quad \text{and} \quad L_t^*(X_{t+1}|X_t) = \log \frac{f_{\theta_1^*}}{f_{\theta_0^*}}(X_{t+1}|X_t),$$

satisfy the following lemma:

**Lemma 3** *The log-likelihood ratio $L_t$ and $L_t^*$ admit the following asymptotic behaviours:*

$$\lim_{\lambda \to \infty} \mathbb{E}_\lambda[L_\lambda^*] < 0 \ \text{ but } \ L_\lambda \xrightarrow[\lambda \to +\infty]{a.s} 0 \ ; \ \lim_{t \to \infty} \mathbb{E}_\lambda[L_t^*|\lambda < \infty] > 0 \ \text{ but } \ L_t \xrightarrow[t \to +\infty]{a.s} 0 \qquad (10)$$

Practically, this means that before the change point, the log likelihood ratio is over-estimated (as it should be smaller than 0), while after the change point, the log likelihood ratio is eventually under-estimated (as it should exceed 0). This is a consequence of Lemma 2. In fact, for $t \gg \lambda$ (respectively for $t \leq \lambda$), both functions $f_0^t$ and $f_1^t$ are approaching $f_1$ (respectively $f_0$). Thus in both cases $\theta_0^t$ and $\theta_1^t$ converge to the same value. However, combined with the behaviour of $f_0^t$ and $f_1^t$ around $\nu_\alpha$ from Lemma 2, we can expect the Kullback–Leibler (KL) divergence $D_{KL}(f_{\theta_0^t}\|f_{\theta_1^t})$ to converge to 0 before and after the change point while peaking around the optimal stopping time. We exploit this observation to mitigate the problems highlighted with Lemma 3.

**Annealing:** The under-estimation of $L_t^*$ after the change point is due to the use of post change observations (drawn from $f_{\theta^*}$) when estimating $\theta_0^*$. In practice, this is particularly problematic when the error rate must be controlled under a low threshold $\alpha$ (equivalently a high cutting threshold $B_\alpha$) or when the pre and post change distributions are very similar. When $t$ approaches the optimal stopping time ($t \approx \nu_\alpha$), the minimising argument of $f_0^t$ is converging to $\theta_0^*$. As $t$ grows, the approximation $f_0^t$ is degraded, while $f_1^t$ is becoming a better approximation, thus $\theta_1^t$ starts converging to $\theta_1^*$. This leads to an increase in the KL divergence. Our proposition is to keep optimising the version of $f_0^t$ that coincides with this increase (which in turn is $f_0^{\nu_\alpha}$: the best candidate to identify $\theta_0^*$). As a fix, we propose: 1- to shift the probability $\mathbb{P}(\boldsymbol{\lambda} > \tau|\nu_\alpha^S = n)$ by replacing it with $\mathbb{P}(\boldsymbol{\lambda} > \tau - \Delta|\nu_\alpha^S = n)$ (where $\Delta$ is the 'delay' of the detection with respect to the Shiryaev stopping time: $(n-\nu_\alpha^S)^+$), and 2- anneal the optimisation steps of $\theta_0^t$ as the KL divergence increases. These tweaks correct the objective function used to learn $\theta_0^*$ and stop its optimisation when the observations start to deviate from the learned pre-change distribution. The delay can be formally introduced by replacing $f_0^t$ with $f_0^{t-\Delta}$ in Equation (9). In practice, $\Delta$ is a delay heuristic that increases as the KL divergence increases. This reduces the noise when learning $\theta_0^*$. The second idea is to use a probability $p_0 = 1 - \Delta\epsilon$ of executing a gradient step when learning the pre-change parameter. This anneals the optimisation of $\theta_0^*$.

**Penalisation:** The over-estimation of $L_t^*$ before the change point, is due to the exclusive use of pre change observation when estimating $\theta_1^*$. This is particularly problematic for application where a high error threshold is tolerable (equivalently a low cutting threshold for the used statistic). This is also an issue when the pre and post change distribution are very different. As a solution, we penalise the log-likelihood using the inverse KL divergence. Before the change point, both parameters converge to $\theta_0^*$: this means that the KL divergence between the estimated distributions is around 0. Inversely, after the change point, if the optimisation of $\theta_0^t$ is annealed, the KL divergence between the estimated parameters must hover around $D_{KL}(f_{\theta_0^*}\|f_{\theta_1^*})$. Formally, penalising the log likelihood can be seen as using $\hat{L}_t$ when computing the statistics $S_n$ where $\hat{L}_t = L_t - c/D_{KL}(f_{\theta_0^t}\|f_{\theta_1^t})$ for some real $c > 0$.

We provide in the appendix an ablation analysis as well as experimental proof of the effectiveness of both annealing and penalisation.

**Algorithm:** In practice, only the first $t$ observations are available in order to evaluate the parameters $(\theta_0^*, \theta_1^*)$. Hence, we design a stochastic gradient descent (SGD) based algorithm to solve Equation (9). Consider the following loss functions, where $I$ is a set of time indices:

$$
\begin{aligned}
\mathcal{L}_0^n(I, \theta_0, \theta_1) &= \sum_{t \in I} \mathbb{P}(\boldsymbol{\tau} < t | \boldsymbol{\tau} \sim f_n^{\theta_0, \theta_1}) \log(\frac{f_{\theta_1}}{f_{\theta_0}})(X_{t+1}|X_t) \\
\mathcal{L}_1^n(I, \theta_0, \theta_1) &= \sum_{t \in I} \mathbb{P}(\boldsymbol{\tau} > t | \boldsymbol{\tau} \sim f_n^{\theta_0, \theta_1}) \log(\frac{f_{\theta_1}}{f_{\theta_0}})(X_{t+1}|X_t)
\end{aligned}
\tag{11}
$$

The quantity $\mathcal{L}_k^t$ is an estimator of $f_k^t$ using the observations $(X_t)_{t \in I}$. The rational previously discussed implies that by re-weighting random samples from the observations we can approximate the functions $f_k$ using the familly of functions $f_k^t$.

A good approximation of $\theta_0^t$ and $\theta_1^t$ can be obtained with few SGD steps. These parameters are used to update the SHIRYAEV or the CUSUM statistic. The change point is declared once the threshold $B_\alpha$ is exceeded. The proposed procedure is described in Algorithm 1. In addition to the classical SGD hyper-parameters (number of epochs, gradient step size, momentum, ...), we propose to control the **penalisation** and the **annealing** procedure using the coefficients $c, \epsilon \geq 0$ respectively.

---

**Algorithm 1** Temporal Weight Redistribution

1: **Procedure:** $\text{TWR}(\theta^0, \theta^1, (X_t)_{t=0}^{t_2}, N_e, B_\alpha, \epsilon, c)$
2: Initialise $S \leftarrow 0$; $\theta_0, \theta_1 \leftarrow \theta^0, \theta^1$; $\Delta \leftarrow 0$; and $p_0 \leftarrow 1$
3: **for** $t \in [1, t_2]$ **do**
4:     **for** $e \in [1, N_e]$ **do**
5:         Randomly sample $I$: a set of indices from $[0, t]$
6:         $\theta_0 \leftarrow$ *gradient update rule*$(\theta_0, \nabla_{\theta_0} \mathcal{L}_0^{t-\Delta}(I, \theta_0, \theta_1))$ with probability $p_0$
7:         $\theta_1 \leftarrow$ *gradient update rule*$(\theta_1, -\nabla_{\theta_1} \mathcal{L}_1^t(I, \theta_0, \theta_1))$
8:     **end for**
9:     $S \leftarrow$ *statistic update rule*$(S, \frac{f_{\theta_1}}{f_{\theta_0}}(X_{t+1}|X_t) - \frac{c}{D_{KL}(f_{\theta_0}|f_{\theta_1})})$
10:    If $S > B_\alpha$ then declare change point and break
11:    If $D_{KL}(f_{\theta_0}|f_{\theta_1}) > \bar{D}$ then $\Delta \leftarrow \Delta + 1$ and $p_0 \leftarrow p_0 - \epsilon$
12:    $\bar{D} \leftarrow \frac{t-1}{t}\bar{D} + \frac{1}{t}D_{KL}(f_{\theta_0}|f_{\theta_1})$
13: **end for**

---

## 4 RELATED WORKS

Existing solutions for the QCD problem under known parameters can be extended to the case of unknown parameters using generalised likelihood ratio (GLR) statistics. For example, an optimal solution for the Lorden formulation is a generalisation of the CUSUM algorithm using the GLR statistic for testing the null hypothesis of no change-point, based on $(X_i)_0^n$, versus the alternative hypothesis of a single change-point prior to $n$. However, given $n$ observation, this solution runs in $\mathcal{O}(n^2)$. Others approached the problem from a Bayesian perspective by keeping track of the run length posterior distribution (the time since the last change point). Since in practice this distribution is highly peaked, the Bayesian Online Change Point Detection (BOCPD) algorithm (Adams & MacKay, 2007) is namely a good alternative in the i.i.d. setting. In fact by pruning out run lengths with probabilities below $\epsilon$ or by considering only the $K$ most probable run lengths, the algorithm runs respectively in $\mathcal{O}(n/\epsilon)$ and $\mathcal{O}(Kn)$ without critical loss of performance. Extensions of BOCPD algorithm to the case of non i.i.d observations, namely the Gaussian Process Change Point Models run in $\mathcal{O}(n^4)$ (Saatchi et al., 2010). If pruning is applied, the complexity can be reduced to $\mathcal{O}(nR^2/\epsilon)$ or $\mathcal{O}(nKR^2)$ where $R$ is the typical unpruned max run length (Saatchi et al., 2010).

To avoid this complexity issue, multiple approximate solutions that run in $\mathcal{O}(1)$ have been proposed over the years. However, a common practice (Nikiforov, 2000; Li et al., 2009; Unnikrishnan et al., 2011; Singamasetty et al., 2017; Molloy & Ford, 2018) is to assume that pre-change parameters are known as this is not a major limitation in many applications (fault detection, quality control...) where the pre-change corresponds to nominal behaviour of a process and can be characterised offline. A sub-optimal solution (Nikiforov, 2000) is to track in parallel $S_n^{\theta_0, \theta_1}$ over a subset of possible post-change distributions and consider the worst case scenario. Somewhat Similarly to our approach,

adaptive algorithms (Li et al., 2009; Singamasetty et al., 2017) attempt to improve this procedure by tracking $S_n^{\theta_0, \bar{\theta}_1}$ where $\bar{\theta}_1 = \arg\max_\theta f_1(\theta_0, \theta)$. Different from proceeding works, our contribution in this paper is an approximate procedure that can handle the case of unknown pre- and post- change parameters, runs in $\mathcal{O}(1)$, and have near optimal performances.

## 5 EXPERIMENTAL RESULTS

**Detection delay on synthetic data:** The distribution family $f_\theta$ used experimentally to simulate data given the pre/post change parameters and the previous observations relies on two deep neural network $(\phi_\mu, \phi_\sigma)$ used to evaluate the parameters of a Gaussian distribution such as: $f_\theta(.|x) = \mathcal{N}(\phi_\mu(\theta, x), \phi_\sigma(\theta, x))$. We compare our algorithm to three other possibilities: the optimal log-likelihood ratio (an oracle having access to the true parameters), the generalised log-likelihood ratio (GLR) and the adaptive log-likelihood ratio where the pre change parameters are learned offline (a fraction of pre-change observations are used to learn $\theta_0$).

We also introduce an alternative metric to the average detection delay: the regretted detection delay. An oracle with access to the true parameters will predict change as fast as the optimal algorithm under known parameters. In order to put performances into perspective, for a given cutting threshold $B$, the regretted average detection delay $\mathcal{R}_B(a)$ of an algorithm, is the additional delay with respect to the oracle. If we denote the oracle's optimal stopping time $\nu_B$ and the algorithm stopping time $\nu_B^a$, then: $\mathcal{R}_B(a) = \mathbb{E}_\lambda[(\nu_B^a - \nu_B)^+ | \nu_B^a \geq \nu_B \geq \lambda]$. This formulation of the regret is conditioned to $\nu_B^a \geq \nu_B$ for the same reason that the measurement of the average detection delay is conditioned to $\nu_B \geq \lambda$. An algorithm that predicts change faster than the optimal one under known parameters, is over-fitting the noise in the data to some extent. For this reason, when comparing performances, there is no true added value in detecting changes faster than the optimal algorithm in the known parameter setting.

We evaluate the performances using the SHIRYAEV update rule for the statistic (solution to the BAYESIAN setting). We provide in the appendix the analysis using the CUSUM update rule (solution to the MIN-MAX setting) with the same conclusions. We evaluate the regretted detection delay in the case where $\mathcal{X} = \mathbb{R}^{10}$, $\Theta = \mathbb{R}^{10}$, and $(\phi_\mu, \phi_\sigma)$ are 5-layer deep, 32-neurons wide neural network. We use a synthetic example where the KL divergence between pre- and post- change distribution is $D_{KL}(f_{\theta_0^*}|f_{\theta_1^*}) = .3$. This allows us to view performances in difficult settings. For higher KL divergence values, change detection becomes easier. We provide in the appendix the same analysis for the case where $D_{KL}(f_{\theta_0^*}|f_{\theta_1^*}) = 3$. In Figure 1, we provide the average regret overs 500 simulations. Each simulated trajectory is 1000 long with a change point at the 500th observation. We use the full pipeline of our algorithm with

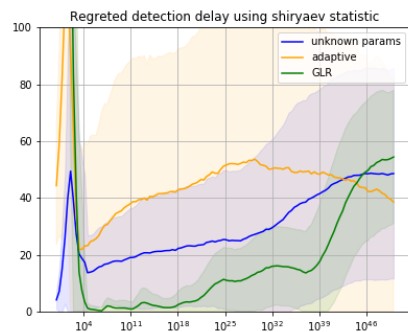

Figure 1: Regret as a function of $B_\alpha$

a penalisation coefficient ($c = 0.01$), and we allow the adaptive algorithm to exploit 10% of the pre change observations. Our algorithm (the blue curve) achieves comparable performance to the GLR statistic (without requiring intensive computational resources) and achieves better and more stable results compared to the adaptive algorithm (without requiring domain knowledge).

**Multi-task reinforcement learning:** Consider a task space $\Theta$ and for each task $\theta \in \Theta$ the associated Markov decision process $\mathcal{M}_\theta = \{\mathcal{S}, \mathcal{A}, \mathcal{P}, \mathcal{R}_\theta, \gamma\}$ where $\mathcal{S}$ is the state space, $\mathcal{A}$ is the action space, $\mathcal{P}$ is the environment dynamics, $\mathcal{R}_\theta$ is the task specific reward function and $\gamma$ is the discount factor. We evaluate algorithms using the copycat agent scenario: a main agent is solving a set of unknown tasks $(\theta_i)_{i=0}^K$ and switches from task to task at unknown change points $(t_i)_{i=0}^K$. We denote $\theta_t^* = \sum_i \theta_i 1_{t_i \leq t < t_{i+1}}(t)$ the task being solved over time. Our goal is to construct a copycat agent predicting the main agent's tasks online through the observation of the generated state action $(s_t, a_t) \in \mathcal{S} \times \mathcal{A}$. This scenario has many different real life applications: Users of any website can be viewed as optimal agents solving iteratively a set of tasks (listening to sets of music genre, looking for different types of clothes, ..). The state space corresponds to the different web-pages and the action space is the set of buttons available to the user. We argue that predicting the task being solved online is an important feature for behaviour forecasting and content recommendation.

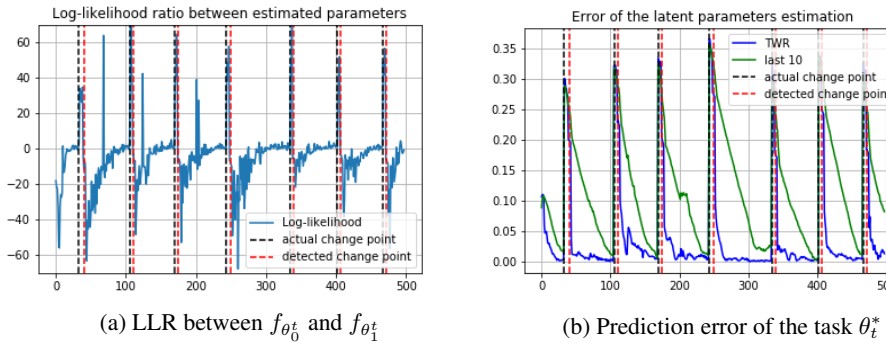

Figure 2: Performance analysis of the copycat problem

In the following let $x_t = (s_t, a_t)$ denote the observations and let $\pi_\theta^*$ denote the optimal policy of $\mathcal{M}_\theta$. The copycat problem is a QCD problem where the observations are drawn from $f_\theta(x_{t+1}|x_t) = \mathcal{P}(s_{t+1}|s_t, a_t)\pi_\theta^*(a_{t+1}|s_{t+1})$. However this implies that we have the optimal policy $\pi_\theta^*$. In practice it is sufficient to either have access to the reward functions $\mathcal{R}_\theta$ or a set of task labelled observations. In both cases the problem of learning $\pi_\theta^*$ is well studied. When $\mathcal{R}_\theta$ is known, then coupling Hindsight Experience Replay (Andrychowicz et al., 2017) with state of the art RL algorithms such as Soft Actor Critic (Haarnoja et al., 2018) can solve a wide range of challenging multi-task problems (Plappert et al., 2018). When a history of observations are available, we can use generative adversarial inverse reinforcement learning (Ho & Ermon, 2016) or Maximum Entropy Inverse Reinforcement Learning (Gleave & Habryka, 2018; Yu et al., 2019) to learn $\pi_\theta^*$.

We evaluate our algorithm performances on the *FetchReach* environment from the *gym* library. Each time the main agent achieves the desired goal, a new task is sampled. We run this experiment for 500 time step. Temporal Weight Redistribution (TWR) is used in order to evaluate the tasks online. Algorithm 1 is easily adapted to the multiple change point setting: each time a change point is detected, we reinitialise the parameters $\Delta$ and $p_0$ and we set the task parameters to the previous post change parameters. As such, we use the running estimate $\theta_0^t$ overtime to approximate the main agent task $\theta_t^*$. Experimental results are reported in Figure 2.

We used the TWR algorithm with an annealing parameter $\epsilon = 0.1$ and a penalisation coefficient $c = 0.01$, and the CUSUM statistic with a cutting threshold $B_\alpha = 50$. We designate the task change points of the main agent with dashed black lines and the detected change points with a dashed red lines. On Figure 2a, we observe that the log-likelihood ratio (LLR) $\log(f_{\theta_1^t}/f_{\theta_0^t})(x_t)$ starts off negative, stabilises around 0 and becomes strictly positive after the actual change point. Once the change detected, we reset the TWR and the LLR falls down to a negative value. This trend is consistently reproduced with each new task. The main agent have a small probability of making a random action, this causes the LLR to peak sometimes before the change point. However this doesn't cause the statistic to exceed the threshold $B_\alpha$. We provide on Figure 2b the estimation error of $\theta_t^*$. The blue curve is the error of the TWR estimate $\|\theta_0^t - \theta_t^*\|$. As a baseline we estimate the task using the last 10 observation: $\hat{\theta}_t = \arg\max_\theta \sum_{i=1}^{10} \log(f_\theta(x_{t-i}))$. The error of this estimator $\|\hat{\theta}_t - \theta_t^*\|$ is plotted in green. The prediction of the TWR copycat agent are clearly more reliable. Additional experimental results for the *Fetchpush* and the *PickAndPlace* environments as well as an analysis of the impact of using learned policies when solving the copycat problem are provided in the appendix.

## 6 CONCLUSION

We studied the QCD problem under unknown parameters for Markovian process. Extending our techniques to handle higher order Markov chains, (i.e., for some window parameter $w > 0$, $f_\theta$ depends of $X_{t-w:t}$ rather than just the last observation) is trivial, thus covering a wide range of natural processes. We provide a tractable algorithm with near optimal performances that relies on the asymptotic behaviour of optimal algorithms in the known parameter case. This mitigates the performance-cost dilemma providing the best of both worlds: a scalable procedure with low detection delay. Empirically, we were able to outperform previous approximate procedure (adaptive algorithms) and approach the performances of prohibitively expensive ones (GLR). Interesting future direction include taking into account the delay caused by the generalisation error.

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

# A    SOLUTION FOR THE QCD PROBLEM UNDER KNOWN PARAMETERS

In this section, we assume that both the parameters $\theta_0$ and $\theta_1$ in Equation (1) are known. We present in the following the main optimality existing results.

**Min-Max formulation:** In the i.i.d. case (i.e. when $f_\theta(.|x) = f_\theta(.)$), the CUSUM algorithm (Lorden et al., 1971) is an optimal solution (Moustakides et al., 1986; Ritov, 1990) for the Lorden formulation (3). However, even though strong optimality results hold, optimising **WADD** is pessimistic. With the delay criterion base on **CADD** (Pollak formulation), algorithms based on the SHIRYAEV-ROBERTS statistic are asymptotically within a constant of the best possible performance (Pollak, 1985; Tartakovsky et al., 2012).
In the non i.i.d. case, state of the art methods (Lai, 1998) are modified versions of the CUSUM algorithm that converge asymptotically to a lower bound of **CADD** (and thus a lower bound to **WADD**).

**Bayesian formulation:** The SHIRYAEV algorithm (Shiryaev, 1963) is asymptotically optimal in the i.i.d. case. This result is extended to the non i.i.d. case under Hypothesis 1 (Tartakovsky & Veeravalli, 2005).

## A.1    ALGORITHMS

In this section, we formally introduce the algorithms used to solve the change point problems under known parameters. A common trait of these algorithms is the computation of a statistic $S_n^{\theta_0,\theta_1}$ and the definition of a cutting threshold $B_\alpha$. The stopping time $\nu_\alpha$ is the first time the statistic exceeds the threshold. The value $B_\alpha$ is chosen such that **PFA** (respectively **FAR**) of the associated stopping time does not exceed the level $\alpha$. In the case of the BAYESIAN formulation, the threshold is simplified into $B_\alpha = \frac{1-\alpha}{\alpha}$.

**The SHIRYAEV Algorithm:**    The general formulation of the statistic $S_n^{\theta_0,\theta_1}$, is the likelihood-ratio of the test of $H_0 : \lambda \leq n$ versus $H_1 : \lambda > n$ given the observations $X_{t \leq n}$. In the general case, with a prior $\mathbb{P}(\boldsymbol{\lambda} = k) = \pi_k$, this statistic writes as:

$$S_n^{\theta_0,\theta_1} := \frac{\sum_{k \leq n} \pi_k \prod_{t=1}^{k-1} f_{\theta_0}(X_{t+1}|X^t) \prod_{t=k}^{n+1} f_{\theta_1}(X_{t+1}|X^t)}{\sum_{k > n} \pi_k \prod_{t=1}^{n-1} f_{\theta_0}(X_{t+1}|X^t)} \tag{12}$$

In the case of geometric prior with parameter $\rho$, the statistic is simplified into:

$$S_n^{\theta_0,\theta_1} = \frac{1}{(1-\rho)^n} \sum_{k=1}^{n} (1-\rho)^{k-1} \prod_{t=k}^{n} L_t(\theta_0,\theta_1) \text{ where } L_t(\theta_0,\theta_1) = \frac{f_{\theta_1}(X_{t+1}|X^t)}{f_{\theta_0}(X_{t+1}|X^t)}$$

The statistic $S_n^{\theta_0,\theta_1}$ can be computed recursively under this simplification:

$$S_0^{\theta_0,\theta_1} = 0 \text{ and } S_{n+1}^{\theta_0,\theta_1} = \frac{1+S_n^{\theta_0,\theta_1}}{1-\rho} L_{n+1}(\theta_0,\theta_1)$$

**The SHIRYAEV-ROBERT Algorithm:**    The statistic to be computed in this case can be seen as an extension of the one in the SHIRYAEV algorithm when $\rho = 0$. The recursive formulation is indeed:

$$S_0^{\theta_0,\theta_1} = 0 \text{ and } S_{n+1}^{\theta_0,\theta_1} = (1 + S_n^{\theta_0,\theta_1}) L_{n+1}(\theta_0,\theta_1)$$

**The CUSUM Algorithm:**    The relevant statistic is defined as: $S_n^{\theta_0,\theta_1} = \max_{k \leq n} \sum_{t=k}^{n} \log(L_t)$, and can be computed recursively by:

$$S_0^{\theta_0,\theta_1} = 0 \text{ and } S_{n+1}^{\theta_0,\theta_1} = (S_n^{\theta_0,\theta_1} + \log(L_{n+1}))^+$$

# B    R-QUICK CONVERGENCE HYPOTHESIS

Hypothesis 1 is not too restrictive, and $r$-quick convergence conditions were previously used to establish the asymptotic optimality of sequential hypothesis tests for general statistical models (Draglia et al., 1999; Lai, 1976; 1981).

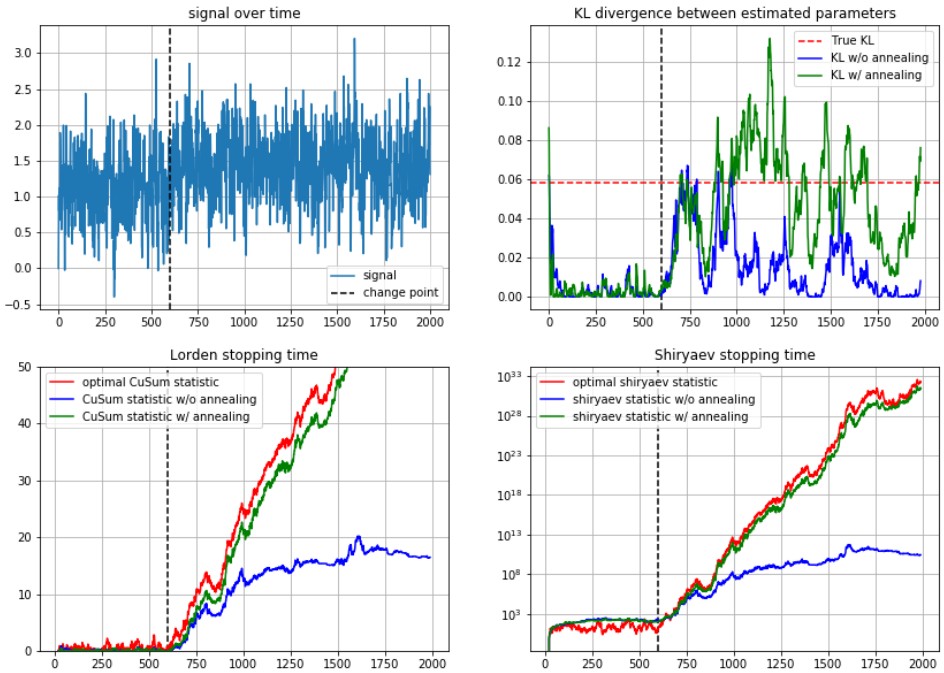

Figure 3: Annealing the optimisation of $\theta_0$

Using the law of large numbers, this hypothesis is satisfied for $q = \mathbb{E}_{X_t \sim f_{\theta_1}} \left[ \frac{f_{\theta_1}(X_t)}{f_{\theta_0}(X_t)} \right]$ in the i.i.d. case. In the (ergodic) Markovian case, the convergence is achieved for irreducible Markov chains due to their ergodicity.

However, the speed of convergence of Equation (4), depends on the speed of convergence to the stationary distribution of the Markov chain. Even though for any $r > 0$ construction of Markov chains that satisfy assumption 4 is possible (Draglia et al., 1999), providing a guarantee of this speed is not a trivial question (Hairer, 2010).

## C  ALGORITHMIC ANALYSIS

### C.1  IMPACT ANALYSIS (ANNEALING AND PENALISATION)

In order to justify experimentally the use of annealing and penalisation to solve the issues highlighted with Lemma 3, we simulate simple examples where the phenomena are amplified to their extreme.

**Annealing:** We consider a one dimensional signal. The KL divergence between the pre and post change distribution is $0.06$. As discussed in the paper, the issue is that after the change point we start using observations from the post change distribution to learn the pre change one. This leads to $\theta_0^t$ and $\theta_1^t$ converging to the post change parameter $\theta_1^*$. In fact, without the annealing procedure, the KL divergence $D_{\mathrm{KL}}(f_{\theta_0^t} \| f_{\theta_1^t})$ (the blue curve, in the top right subplot in Figure 3) ends up collapsing to $0$. This has the problematic consequence of slowing down the statistics. In fact, both the SHIRYAEV and the CUSUM statistics (the blue curves in the lower subplots of Figure 3) start adjacent to the optimal statistic (in red) and slowly degrade in quality. On the other hand, implementing the annealing procedure -as described in the paper with a step size $\epsilon = 0.01$- solves this issue. The KL divergence (green curve) hovers around the true value and the computed statistics are almost within a constant of the optimal one.

**Penalising:** We consider in this case a one dimensional signal with a KL divergence of $2$. The problem we are analysing in this setting is the exclusive use of pre change observations when learning the parameters before the change point. The optimal log-likelihood ratio (LLR) before the change point is $-2$ (the red curve in the top right subplot of Figure 4) while the learned one (in blue) is

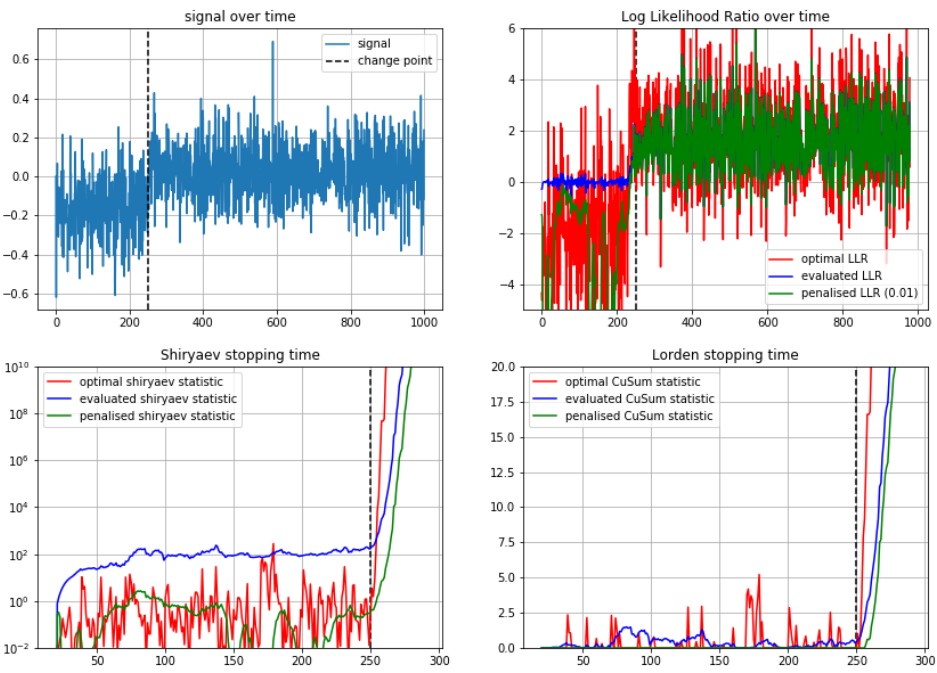

Figure 4: Penalising the Log Likelihood Ratio

around 0. This is due to $\theta_0^t$ and $\theta_1^t$ converging to the pre change parameter $\theta_0^*$. This over-estimation of the LLR leads to a higher estimate of the SHIRYAEV statistic. In the lower left subplot of Figure 4, we have an increase of 100 fold compared to the optimal value. Penalising the LLR, using a coefficient $c = 0.01$, solves this issue without being detrimental to the post change performance. In fact the penalised LLR (the green curve) is strictly negative before the change point and has a similar values to the optimal ones afterwards. The associated SHIRYAEV statistic (the green curve) is less prone to detect false positives as it has the same behaviour as the optimal one. This safety break (penalisation) comes with a slight drawback for higher cutting thresholds as it induces an additional delay due to the penalising component. Interestingly, the CUSUM statistic does not have the same behaviour in this example. This is explained with the nature of the LORDEN formulation. As it minimises a pessimistic evaluation of the delay (**WADD**), the impact of a LLR of $0$ is less visible. However, on average, the same phenomena occurs. This will be observed in the ablation analysis, where on average, there was no notable differences between the different formulations.

## C.2 ABLATION ANALYSIS (ANNEALING AND PENALISATION)

Since the algorithm integrates different ideas to mitigate the issues discussed in the paper, an ablation study is conducted to understand their contribution. We simulate the case where $\mathcal{X} = \mathbb{R}^{10}, \Theta = \mathbb{R}^{10}$, and $(\phi_\mu, \phi_\sigma)$ are 5-layer deep, 32-neurons wide neural network. The penalisation coefficient is fixed to $0.01$ and the annealing step is fixed to $0.02$. In Figure 5, the regretted detection delay is evaluated using $500$ simulations. We consider a mid-ground complexity case, with a KL divergence of $1.5$, which is a more realistic scenario in real life situations.

We provide the regret with respect to both the optimal performance under known parameters for both the BAYESIAN setting (SHIRYAEV statistic on the left) and the MIN-MAX setting (CUSUM statistic on the right). We split the analysis of the curves to three scenarios: The low, the mid and the high cutting threshold cases. They correspond respectively to 3 categories of risk inclinations levels: the high, the mid and the low.

**The high risk inclination:** In the presented case, this correspond to a cutting threshold $B$ smaller than $10^{20}$ in the shiryaev algorithm and than $50$ in the CUSUM algorithm. This can be associated to household applications and relatively simple settings with high reaction time. In this case, the

simplest version of the algorithm is sufficient to achieve relatively small regret values. In fact, the green curve (associated to the configuration without annealing) presents a relatively low regret with respect to the optimal delay.

**The low risk inclination:** In the presented case, this correspond to $B$ higher than $10^{50}$ in the shiryaev algorithm and than 100 in the cusum algorithm. This can be associated to critical applications where precision outweighs the benefice of speed such as energy production. In this case, configurations without annealing (presented in green here) have poor performance. The best configuration is to perform annealing but without penalisation of the log likelihood.

**The intermediate risk inclination:** This coincides with the remaining spectrum of cases. For example, when the application is critical but requires high reaction time, or when the application has no reaction time constraints but doesn't require extreme safety measures. In this setting, we observe that the best performance require some amount of penalisation of the log likelihood. The choice of the coefficient should be guided by the KL divergence between the pre and post change distributions, with $c = 0$ when the change is subtle (small KL).

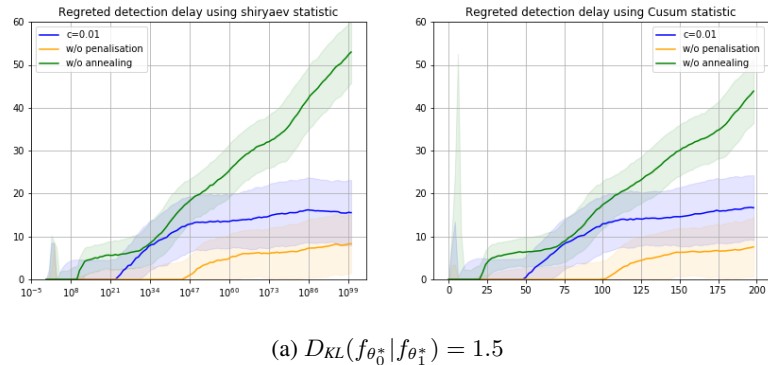

(a) $D_{KL}(f_{\theta_0^*}|f_{\theta_1^*}) = 1.5$

Figure 5: Ablation analysis of the regret detection delay as a function of the cutting threshold

## C.3 Divergence measures based algorithmic variant

In order to solve the QCD problem under unknown parameters, we propose in this paper to approximate the log-likelihood ratio $L_t^*$ efficiently. We achieved this by optimising a surrogate of the KL divergence between the estimated parameters $(\theta_t^0, \theta_t^1)$ and the true parameters $(\theta_0^*, \theta_1^*)$. Algorithm 1 devises a theoretically grounded weighting technique that approximates expectations under pre/post change distributions. For this reason, other divergence measure based loss functions can be used instead of the proposed ones in Equation 11. In particular, consider the following loss functions:

$$\mathcal{L}_0^n(I, \theta_0, \theta_1, g) = \sum_{t \in I} \mathbb{P}(\tau < t | \tau \sim f_n^{\theta_0, \theta_1}) g(\tfrac{f_{\theta_1}}{f_{\theta_0}})(X_{t+1}|X_t)$$
$$\mathcal{L}_1^n(I, \theta_0, \theta_1, g) = \sum_{t \in I} \mathbb{P}(\tau > t | \tau \sim f_n^{\theta_0, \theta_1}) g(\tfrac{f_{\theta_1}}{f_{\theta_0}})(X_{t+1}|X_t)$$

For $k \in \{0, 1\}$, the loss function $\mathcal{L}_k^n(I, \theta_0, \theta_1, g)$ approximates $\mathbb{E}_k\left[g(\tfrac{f_{\theta_1}}{f_{\theta_0}})\right]$. Under the assumptions that $g$ is an increasing function with $g(1) = 0$ and $h(x) = x.g(x)$ is convex, $\mathcal{L}_0^n$ becomes a proxy of the $h$-divergence measure $D_h(f_{\theta_1^*}\|f_{\theta_0^*})$ (respectively $D_h(f_{\theta_0^*}\|f_{\theta_1^*})$ for $\mathcal{L}_1^n$). This formulation is coherent with the QCD objective as it minimises $L_t$ under the pre-change distribution and maximises it under the post change distribution and we can use it as a valid change point detection algorithm. The obtained parameters $(\theta_t^0, \theta_t^1)$ are the furthest apart with respect to the associated h-divergence. The KL divergence (associated with $g(x) = \log(x)$) is a natural fit for our setting as it simultaneously ensures that the evaluated parameters converge to $(\theta_0^*, \theta_1^*)$.

We keep using the same hyper-parameters from the ablation analysis in order to asses the **ADD** of the TWR algorithm using different divergence measures. The experimental results are reported in Figure 6. In addition to the KL-based version (dark blue curve), we consider the case where $g(x) = \sqrt{x} - 1$ (purple curve) and the case where $g(x) = (x - 1).\log(x)$ (cyan curve). We also provide the adaptive (yellow) and GLR (green) average detection delay as a baseline.

The KL-divergence based approach provides comparable **ADD** to the GLR, outperforming both the adaptive algorithm and the other variant of the TWR.

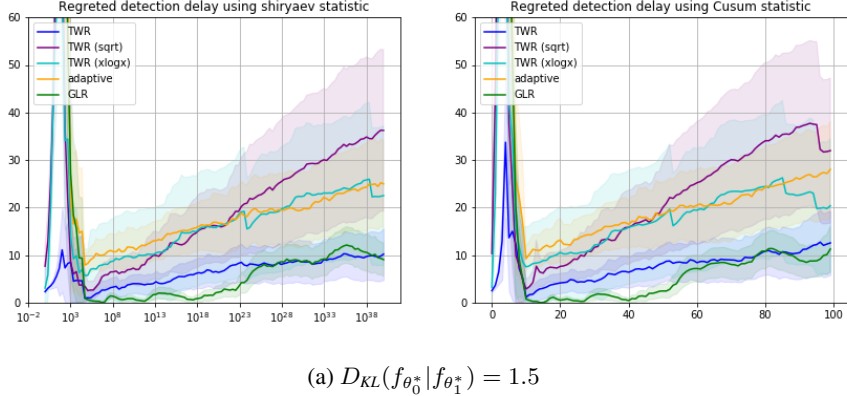

(a) $D_{KL}(f_{\theta_0^*}|f_{\theta_1^*}) = 1.5$

Figure 6: Regretted average detection delay using different divergence measures

## D    PROOFS OF TECHNICAL RESULTS

**Proof of Lemma 1:**    We start by observing the following:

$$
\begin{cases}
f_0(\theta, \theta_1) = \mathbb{E}_0\big[\log(\frac{f_{\theta_1}}{f_{\theta_0^*}})\big] + \mathbb{E}_0\big[\log(\frac{f_{\theta_0^*}}{f_\theta})\big] \\
f_1(\theta_0, \theta) = \mathbb{E}_1\big[\log(\frac{f_{\theta_1^*}}{f_{\theta_0}})\big] - \mathbb{E}_1\big[\log(\frac{f_{\theta_1^*}}{f_\theta})\big]
\end{cases}
$$

Notice that both quantities $\mathbb{E}_0\big[\log(\frac{f_{\theta_1}}{f_{\theta_0^*}})\big]$ and $\mathbb{E}_1\big[\log(\frac{f_{\theta_1^*}}{f_{\theta_0}})\big]$ are constants with respect to the parameter $\theta$, and that:

$$
\begin{cases}
\mathbb{E}_0\big[\log(\frac{f_{\theta_0^*}}{f_\theta})\big] = D_{KL}(f_{\theta_0^*}|f_\theta) \\
\mathbb{E}_1\big[\log(\frac{f_{\theta_1^*}}{f_\theta})\big] = D_{KL}(f_{\theta_1^*}|f_\theta)
\end{cases}
$$

where $D_{KL}$ is the Kullback–Leibler divergence in the i.i.d case and the Kullback–Leibler divergence rate in the Markov chains case, i.e.

$$
D_{KL}(f_{\theta_0}|f_{\theta_1}) = \begin{cases}
\int_x f_{\theta_0}(x) \log(\frac{f_{\theta_0}}{f_{\theta_1}})(x) \\
\int_x \Pi_0(x) \int_y f_{\theta_0}(y|x) \log(\frac{f_{\theta_0}}{f_{\theta_1}})(y|x)
\end{cases}
$$

When well defined, $D_{KL}(f_{\theta_0}|f_{\theta_1})$ is strictly positive except when $f_{\theta_0} = f_{\theta_1}$. This allows us to conclude that:

$$
\begin{cases}
\arg\min_\theta f_0(\theta, \theta_1) = \arg\min_\theta D_{KL}(f_{\theta_0^*}|f_\theta) = \theta_0^* \\
\arg\max_\theta f_1(\theta_0, \theta) = \arg\max_\theta -D_{KL}(f_{\theta_1^*}|f_\theta) = \theta_1^*
\end{cases}
$$

This concludes the proof of the Lemma.

**Proof of Lemma 2:**    We denote $\mu_0$ the initial distribution according to which $X_0$ is sampled and $\mu_t$ the distribution of $X_t$. We re-write the functions $f_k^n$ as:

$$
\begin{aligned}
f_0^n(\theta_0, \theta_1) &= \sum_{t=0}^{t_2} \frac{\mathbb{P}(t<\boldsymbol{\lambda}|\nu_\alpha = n)}{\sum_{i=0}^{t_2} \mathbb{P}(i<\boldsymbol{\lambda}|\nu_\alpha = n)} \mathbb{E}_{0,t}[\log(\frac{f_{\theta_1}}{f_{\theta_0}})(X|X_p)] \\
f_1^n(\theta_0, \theta_1) &= \sum_{t=0}^{t_2} \frac{\mathbb{P}(t>\boldsymbol{\lambda}|\nu_\alpha = n)}{\sum_{i=0}^{t_2} \mathbb{P}(i>\boldsymbol{\lambda}|\nu_\alpha = n)} \mathbb{E}_{1,t}[\log(\frac{f_{\theta_1}}{f_{\theta_0}})(X|X_p)]
\end{aligned}
$$

where $\mathbb{E}_{k,t}$ is taken with respect to $(X_p, X) \sim \mu_t(X_p) \times (f_{\theta_0^*}(X|X_p)\mathbb{1}_{t<\lambda} + f_{\theta_1^*}(X|X_p)\mathbb{1}_{t>\lambda})$.
It's important to notice for what follows that $\mathbb{P}(t < \boldsymbol{\lambda}|\nu_\alpha = n)$ (respectively $\mathbb{P}(t > \boldsymbol{\lambda}|\nu_\alpha = n)$) is

decreasing (increasing) with respect to $t$. We also highlight that for any finite set $I$, we have the following:

$$\sum_{t \in I} \frac{\mathbb{P}(t < \boldsymbol{\lambda} | \nu_\alpha = n)}{\sum_{i=0}^{t_2} \mathbb{P}(i < \boldsymbol{\lambda} | \nu_\alpha = n)} \xrightarrow{n, t_2 \to \infty} 0$$
$$\sum_{t \in I} \frac{\mathbb{P}(t > \boldsymbol{\lambda} | \nu_\alpha = n)}{\sum_{i=0}^{t_2} \mathbb{P}(i > \boldsymbol{\lambda} | \nu_\alpha = n)} \xrightarrow{n, t_2 \to \infty} 0$$

Consider the case where $\lambda = \infty$. As $X_t$ is a irreducible Markov chain, we have that

$$||\Pi_0^* - \mu_t||_{\mathbf{T.V}} \xrightarrow{t \to \infty} 0$$

where $||.||_{\mathbf{T.V}}$ is the total variation norm. This means that for any small value $\epsilon > 0$, there exist $T > 0$ such that:

$$\forall t \geq T \quad ||\Pi_0^* - \mu_t||_{\mathbf{T.V}} \leq \epsilon$$

Thus, as the observation $X_t$ are bounded, we have for any continuous function $f$, there exist $T_\epsilon > 0$ such that:

$$\forall t \geq T_\epsilon \quad |\mathbb{E}_{k,t}[f(X | X_p)] - \mathbb{E}_0[f(X | X_p)]| \leq \epsilon$$

This implies that when $\lambda = \infty$ then $f_0^\infty$ and $f_1^\infty$ converges to $f_0$. In fact for any $\epsilon$, there exist $N$ and $T_2$ such that for all $n > N$:

$$\forall t_2 \geq T_2 \qquad \sum_{t < T_\epsilon} \frac{\mathbb{P}(t > \boldsymbol{\lambda} | \nu_\alpha = n)}{\sum_{i=0}^{t_2} \mathbb{P}(i > \boldsymbol{\lambda} | \nu_\alpha = n)} \leq \epsilon$$
$$\forall t_2 \geq T_2 \qquad \sum_{t < T_\epsilon} \frac{\mathbb{P}(t < \boldsymbol{\lambda} | \nu_\alpha = n)}{\sum_{i=0}^{t_2} \mathbb{P}(i < \boldsymbol{\lambda} | \nu_\alpha = n)} \leq \epsilon$$
$$\forall t_2 \geq t \geq T_\epsilon \quad |\mathbb{E}_{k,t}[f(X | X_p)] - \mathbb{E}_0[f(X | X_p)]| \leq \epsilon$$

The case where $\lambda$ is finite can be deduced by considering the sequence $X_{t > \lambda}$. All the observation are sampled according to $f_{\theta_1^*}$, and thus $f_0^\infty$ and $f_1^\infty$ converges to $f_1$.

This result is also valid for $f_1^{\nu_\alpha}$. In fact, as $\mathbb{P}(t > \lambda | \nu_\alpha = n)$ is increasing with respect to $t$, then for any fixed horizon $H$ we have:

$$\sum_{t < H} \frac{\mathbb{P}(t > \boldsymbol{\lambda} | \nu_\alpha = n)}{\sum_{i=0}^{t_2} \mathbb{P}(i > \boldsymbol{\lambda} | \nu_\alpha = n)} \xrightarrow{t_2 \to \infty} 0.$$

As this remains true for $H = T_\epsilon$, we obtain that $f_1^{\nu_\alpha}$ converges to $f_1$.

For a given integer $n$, if $t_2 = \lambda + n$, the convergence of $f_0^{\nu_\alpha}$ to $f_0$ is achieved as $\lambda \to \infty$. In order to establish this result, we need to prove that $\sum_{t = \lambda}^{t_2} \frac{\mathbb{P}(t < \boldsymbol{\lambda} | \nu_\alpha)}{\sum_{i=0}^{t_2} \mathbb{P}(i < \boldsymbol{\lambda} | \nu_\alpha)}$ is decreasing with respect to $\lambda$. This is a consequence of $\mathbb{P}(t < \boldsymbol{\lambda} | \nu_\alpha)$ being decreasing with respect to $t$. In fact:

$$\sum_{t = \lambda}^{t_2} \mathbb{P}(t < \boldsymbol{\lambda} | \nu_\alpha) \leq \sum_{t = \lambda}^{t_2} \mathbb{P}(\lambda < \boldsymbol{\lambda} | \nu_\alpha) = n \mathbb{P}(\lambda < \boldsymbol{\lambda} | \nu_\alpha)$$
$$\sum_{t = 0}^{t_2} \mathbb{P}(t < \boldsymbol{\lambda} | \nu_\alpha) \geq \sum_{t = 0}^{\lambda} \mathbb{P}(\lambda < \boldsymbol{\lambda} | \nu_\alpha) = \lambda \mathbb{P}(\lambda < \boldsymbol{\lambda} | \nu_\alpha)$$

as a consequence the following inequality holds:

$$\sum_{t = \lambda}^{t_2} \frac{\mathbb{P}(t < \boldsymbol{\lambda} | \nu_\alpha)}{\sum_{i=0}^{t_2} \mathbb{P}(i < \boldsymbol{\lambda} | \nu_\alpha)} \leq \frac{n}{\lambda} \xrightarrow{\lambda \to \infty} 0.$$

As such, for any $\epsilon > 0$, there exists $A_n > 0$ such that:

$$\forall \lambda \geq A_n \qquad \sum_{t = \lambda}^{t_2} \frac{\mathbb{P}(t < \boldsymbol{\lambda} | \nu_\alpha)}{\sum_{i=0}^{t_2} \mathbb{P}(i < \boldsymbol{\lambda} | \nu_\alpha)} \leq \epsilon$$
$$\forall \lambda \geq T_\epsilon \qquad \sum_{t = 0}^{T_\epsilon} \frac{\mathbb{P}(t < \boldsymbol{\lambda} | \nu_\alpha)}{\sum_{i=0}^{t_2} \mathbb{P}(i < \boldsymbol{\lambda} | \nu_\alpha)} \leq \epsilon$$
$$\forall \lambda \geq t \geq T_\epsilon \quad |\mathbb{E}_{0,t}[f(X | X_p)] - \mathbb{E}_0[f(X | X_p)]| \leq \epsilon$$

**Proof of Lemma 3:** We have $f_k^\lambda$ converges to $f_0$ as $\lambda \to \infty$ (respectively $f_k^t$ converges to $f_1$ as $t \to \infty$). Given that $\theta_0^*$ minimises $f_0$ (respectively $\theta_1^*$ minimises $f_1$), then $\theta_k^\lambda$ and $\theta_k^t$ satisfy the following:

$$\theta_0^\lambda, \theta_1^\lambda \xrightarrow[\lambda \to +\infty]{a.s} \theta_0^* \quad \text{and} \quad \theta_0^t, \theta_1^t \xrightarrow[t \to +\infty]{a.s} \theta_1^*.$$

As such, we obtain the following result:

$$L_\lambda \xrightarrow[\lambda \to +\infty]{a.s} 0 \quad \text{and} \quad L_t \xrightarrow[t \to +\infty]{a.s} 0$$

On the other hand we have the following:

$$\lim_{\lambda \to \infty} \mathbb{E}_\lambda[L_\lambda^*] = \mathbb{E}_{X_{\lambda-1} \sim \Pi_0^*, X_\lambda \sim f_{\theta_0^*}}[L_\lambda^*] = -D_{K.L}(f_{\theta_0^*} \| f_{\theta_1^*}) \quad < 0$$
$$\lim_{t \to \infty} \mathbb{E}_\lambda[L_t^*] = \mathbb{E}_{X_{t-1} \sim \Pi_1^*, X_t \sim f_{\theta_1^*}}[L_\lambda^*] = D_{K.L}(f_{\theta_0^*} \| f_{\theta_1^*}) \quad > 0$$

This concludes the proof of the lemma.

# E   ADDITIONAL EXPERIMENTAL RESULTS

## E.1   HYPER-PARAMETER SELECTION

The algorithm we designed requires the selection of a few hyper-parameters in order to run properly. In this section we address the issue of tuning them.

The first set of parameters are for optimisation purpose, and thus we advise selecting them according to the complexity of the probability distribution family $\{f_\theta, \theta \in \Theta\}$. In all the experimental settings we used a batch size of 32, a number of epoch equal to 25 and a gradient step size of 0.001. The initial task parameters $\theta^0$ and $\theta^1$ are chosen randomly unless stated otherwise.

As for the cutting threshold, the penalisation and the annealing coefficients, they depend on the KL divergence between the pre- and post- change distribution (thus the task sampling distribution $\mathcal{F}$) and on the mixing times of these distribution. Fine tuning using grid search is the most efficient way to identify suitable candidates.

From our experimental inquiries, we advise a low penalisation coefficient (e.g. $c = 0.001$) unless the threshold $B_\alpha$ is smaller than 100 for the SHIRYAEV statistic and smaller than 10 for the CUSUM statistic. The annealing parameter $\epsilon$ depends on the mixing time of the Markov chain. In fact $\epsilon$ reflects to what extent we keep learning the pre-change parameter once it's similar to the post-change one. In practice we found out that a small coefficient (e.g. $\epsilon = 0.01$) is advised when the observations converge slowly to the stationary distribution. On the other hand, when convergence occurs in few observations, a higher coefficient is a safer choice.

## E.2   COPYCAT AGENT PROBLEM: COMPLEMENTARY RESULTS

In this section we further the analysis of the copycat agent problem. The *Fetch* set of environments provides a good mix of complexity and interpretability. The ability of the TWR algorithm to detect changes and to solve the copycat agent problem in these environments is proof of it's robustness.

### SHARED BEHAVIOURS

When different tasks are associated to a shared behavior, it becomes difficult to evaluate the parameter $\theta_t^*$. This is true for the *push* and the *PickandPlace* environments. In both cases, the goal is to control a robotic hand (through its joints' movements) in order to interact with an object until it reaches a final position (either push it there or pick it and then place it there).

The task space in these problems is the set of possible final positions. No matter what the task is, the main agent is going to reach to the object. The observations associated to this intermediate process of reaching are probably going to yield a poor estimation of the final position. However as the agent start to interact with the object, it starts to become clear what task is being executed. For this reason, an artificial change is detected between the reaching behaviour and the interaction one when using the TWR algorithm. Even though this detection is unwanted (in the sense that there was no actual change of behavior) it allows to reduce the estimation error $\|\theta_0^t - \theta_t^*\|$.

In this section, the experiment we run consists of performing three randomly sampled tasks, each over a window of 20 observations. We estimate the running task $\theta_t^*$ using the TWR algorithm and a maximum log-likelihood estimator (MLE) over both the last 5 and 10 observations. We average the errors over 500 trajectories and report the results in Figure 7. In both environments estimating the task with the running pre-change parameter of the TWR algorithm outperforms MLE. The gap grows narrower however in the *PickAndPlace* setting as the shared behaviour last longer. This makes estimations more difficult.

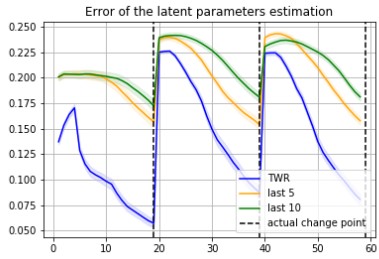 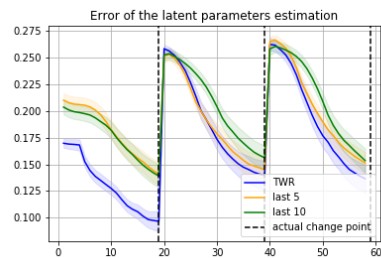

(a) Prediction error in the Push problem      (b) Prediction error in the Pick&Place problem

Figure 7: Performance analysis of the copycat problem

We also notice that the first task estimation of TWR is better than the MLE estimator despite the fact that all the observations are generated with a single parameter. This strange result is due to the detection of an artificial change after the "shared" behaviour. optimising only observations that are clearly correlated to a particular task, leads to better estimations. As for the gap in the first estimation, this is explained by a built-in agnostic behaviour in the TWR algorithm. When the number of observation is less than the **ADD** of the SHIRYAEV algorithm, the pre-change weight term $\mathbb{P}(\boldsymbol{\tau} < t | \boldsymbol{\tau} \sim f_n^{\theta_0^t, \theta_1^t})$ of $\mathcal{L}_0^n$ is extremely small. and thus the estimation remains close to the initialisation. In addition, the tow target functions $\mathcal{L}_0^t$ and $\mathcal{L}_1^t$ are adversarial as they kind of optimise the opposite of each other. This leads to smaller gradient updates. On the other hand, MLE tend to commit to a particular estimate using observations from the 'shared' behaviour phase. This leads generally to a heavily over-fitting estimate.

The agnostic behaviour is more pronounced in Figure 7b (*PickAndPlace*) as the robotic hand picks the object exactly the same way no mater what task is being solved. As long as there is few evidence of a distinctive behaviour occurring, the TWR estimate remains close to the initialisation. In fact the tasks here are positions and the error is the distance between the true and predicted coordinates. In the *PickAndPlace* scenario, the target is randomly sampled in a cube of edge length $0.3$, in the *Push* scenario, it is sampled in a square with the same edge length. with random initialisation we get an average error of $0.1 \times \sqrt{3}$ and $0.1 \times \sqrt{2}$ respectively with a random initialisation. This coincide with the TWR error (blue curve) in both environments given few observations proving that the learned task remained close to the initialisation.

In Figure 7a, the improvement due to the artificial detection of change between the shared behaviour and the task specific one is marked around the $5^{th}$ observation. In fact once the robotic hand starts pushing the object in a particular way, the estimation error starts to decay faster.

PERFORMANCE COST OF LEARNING THE POLICY

Estimating the running task $\theta_t^*$ in the copycat problem with either the TWR algorithm or the MLE requires access to the probability distribution $\pi_\theta^*$. However, in real life situations, we can only learn an approximation of this distribution through historical observations.

In this section we evaluate experimentally the impact of using inverse reinforcement learning to construct an approximate policy $\hat{\pi}_\theta$ and using it as a substitute for $\pi_\theta^*$. We consider the *Reach* environment where the task is to move the robotic hand to a particular position. The main agent execute 5 different tasks, sampled randomly, each over a window of 20 time steps. We evaluate the running task with TWR and MLE using the last 5 observations using the actual policy $\pi_\theta^*$ and the learned one $\hat{\pi}_\theta$. We average the estimation error over 100 simulations and we provide the experimental results in Figure 8. The approximate policy is constructed using the GAIL algorithm and a data-set of task labelled observations generated using $\pi_\theta^*$.

As established before, the TWR estimates using the true policy (in blue) outperform MLE based approaches. This trends remains true on average when using $\hat{\pi}_\theta$. However, the true policy based MLE (in green) converges to a better estimation in the last observations before the change point, while the GAIL based TWR estimate (in red) as well as the GAIL based MLE (in orange) seem to be unable

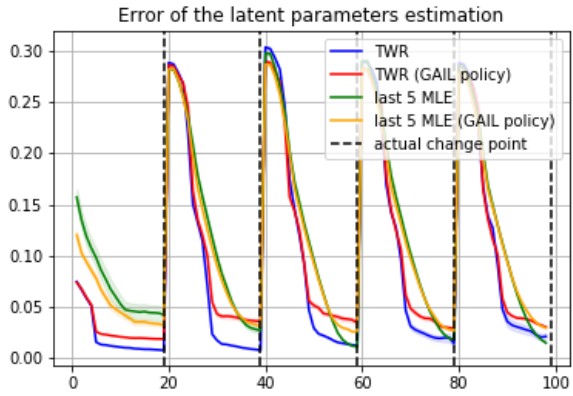

Figure 8: Impact of learning the policy

to improve. This is the cost of using a policy approximation. The additional error is built in due to estimation error of the learned policy.

### E.3 COMPLEMENTARY PERFORMANCE ANALYSIS

In this section we provide complementary experimental results to the ones introduced in the paper. We keep using the same hyper-parameters (*i.e.* $\mathcal{X} = \mathbb{R}^{10}$, $\Theta = \mathbb{R}^{10}$, and $(\phi_\mu, \phi_\sigma)$ are 5-layer deep, 32-neurons wide neural network).

#### SINGLE CHANGE POINT PERFORMANCE

In Figure 9, we provide the average regret overs 500 simulations in both the BAYESIAN (on the left) and the MIN-MAX (on the right) settings. Each simulated trajectory is 1000 observation long with a change point at the 500th observation. We use the full pipeline of our algorithm with a penalisation coefficient ($c = 0.01$), and we allow the adaptive algorithm to exploit $10\%$ of the pre change observations. Our algorithm (the blue curve) achieves comparable performance to the GLR statistic (without requiring intensive computational resources) and achieves better and more stable results compared to the adaptive algorithm (without requiring domain knowledge).

In all experiments and for all considered change detection algorithms, we observe a spike and a high variance of the average detection delay for low cutting threshold. This is a direct consequence of the definition of $B_\alpha$. In fact a low cutting threshold implies a high tolerance of the type I error. This implies that the optimal detection time is no longer a reliable estimate.

#### MULTIPLE CHANGE POINTS PERFORMANCE

We reconsider in this section the multiple change points problem introduced early on in the paper. If the change points are sufficiently separated (i.e. $t_k - t_{k-1}$ is big enough), they can be seen as independent single change point problems. However there is no good reason to believe that this is the case of real life applications. For this reason, the detection delay of the first change point $t_1$ will probably reduce the accuracy at which we estimate the parameter $\theta_1$, this in turn will affect the accuracy of estimating the log-likelihood for the next change point $t_2$. This behaviour will keep on snowballing until we reach a breaking point at which we will miss a change point. Some attempts to deal with this issue have been made in the past to improve the convergence rate of the GLR algorithm in order to reduce the impact of this problem (Ross, 2014). These approaches remain computationally extensive and thus are not considered in our comparison setting.

We provide however an analysis of the impact of the number of pre change observations on the average detection delay in the single change point setting. We keep using the same hyper-parameters where $\mathcal{X} = \mathbb{R}^{10}$, $\Theta = \mathbb{R}^{10}$, and $(\phi_\mu, \phi_\sigma)$ are 5-layer deep, 32-neurons wide neural network. We use parameters achieving a KL divergence of $1.5$ between the pre and post change parameters, and

evaluate the average detection delay over 500 simulations where $t_2 = \lambda + 250$. By varying $\lambda$ we simulate different multiple change point scenarios where the accumulated delay exhausts most of the available observations. The first thing to notice is that our algorithms (in the blue curve) has similar average detection delay to the GLR (green curve) in all presented cases. When we have sufficient observations ($\lambda = 250$), all algorithms will have reasonable performances compared to the optimal delay. However, for smaller horizon, the adaptive algorithm (the orange curve) is predicting change way before the optimal algorithm. This is explained with the bad estimation of the pre change distribution due to a reduced sample size. In fact for cutting threshold up to $10^{25}$, the adaptive algorithm is predicting the change point before it happens. In addition, we observe that our approach varies less than the adaptive one as it's less prone to fit statistical fluctuations.

We provide an analysis of the impact of the number of pre change observations on the average detection delay in both the BAYESIAN (on the left) and the MIN-MAX (on the right) settings. We use parameters achieving a KL divergence of 1.5 between the pre and post change parameters, and evaluate the average detection delay over 500 simulations where $t_2 = \lambda + 250$. By varying $\lambda$ we simulate different multiple change point scenarios where the accumulated delay exhausts most of the available observations. There is no notable variation in the algorithms behaviour when using either the SHIRYAEV or the CUSUM statistic. As for the impact of the number of available pre-change observation before the change point ($\lambda$), the results presented in Figure 10 illustrate how our algorithm is less prone than adaptive procedures to over fit the observation when their number is limited. In fact the blue curve (our algorithm) is closer than the adaptive one (orange curve) to the GLR (green curve) performances. All approaches end up having a constant regret with respect to the optimal detection given enough observations ($\lambda = 250$).

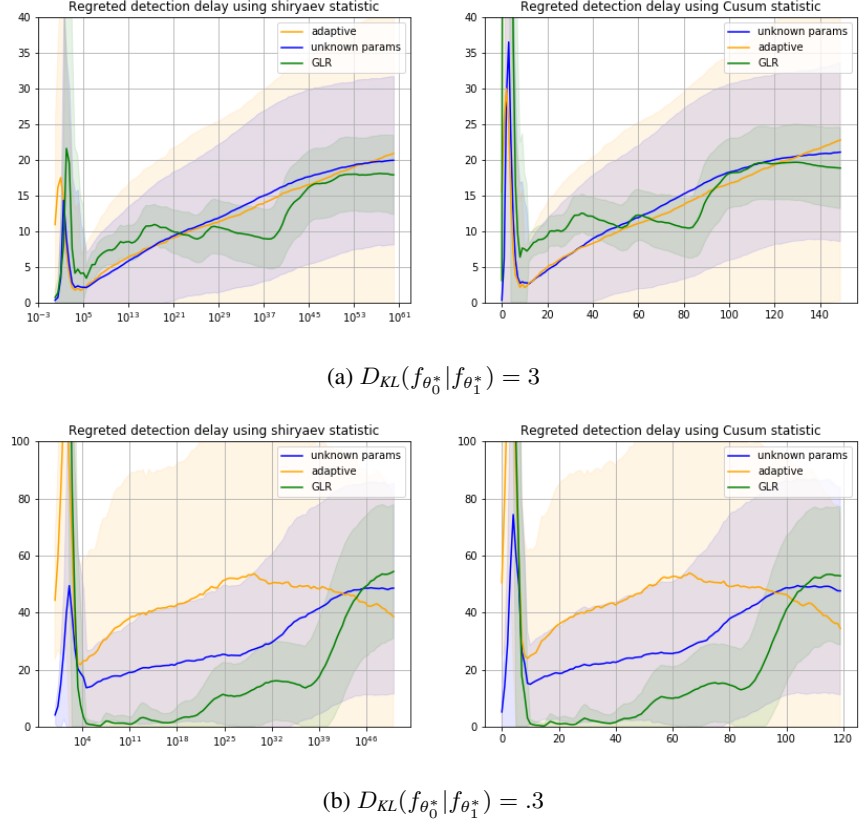

(a) $D_{KL}(f_{\theta_0^*}|f_{\theta_1^*}) = 3$

(b) $D_{KL}(f_{\theta_0^*}|f_{\theta_1^*}) = .3$

Figure 9: Performance analysis of the regretted detection delay as a function of the cutting threshold

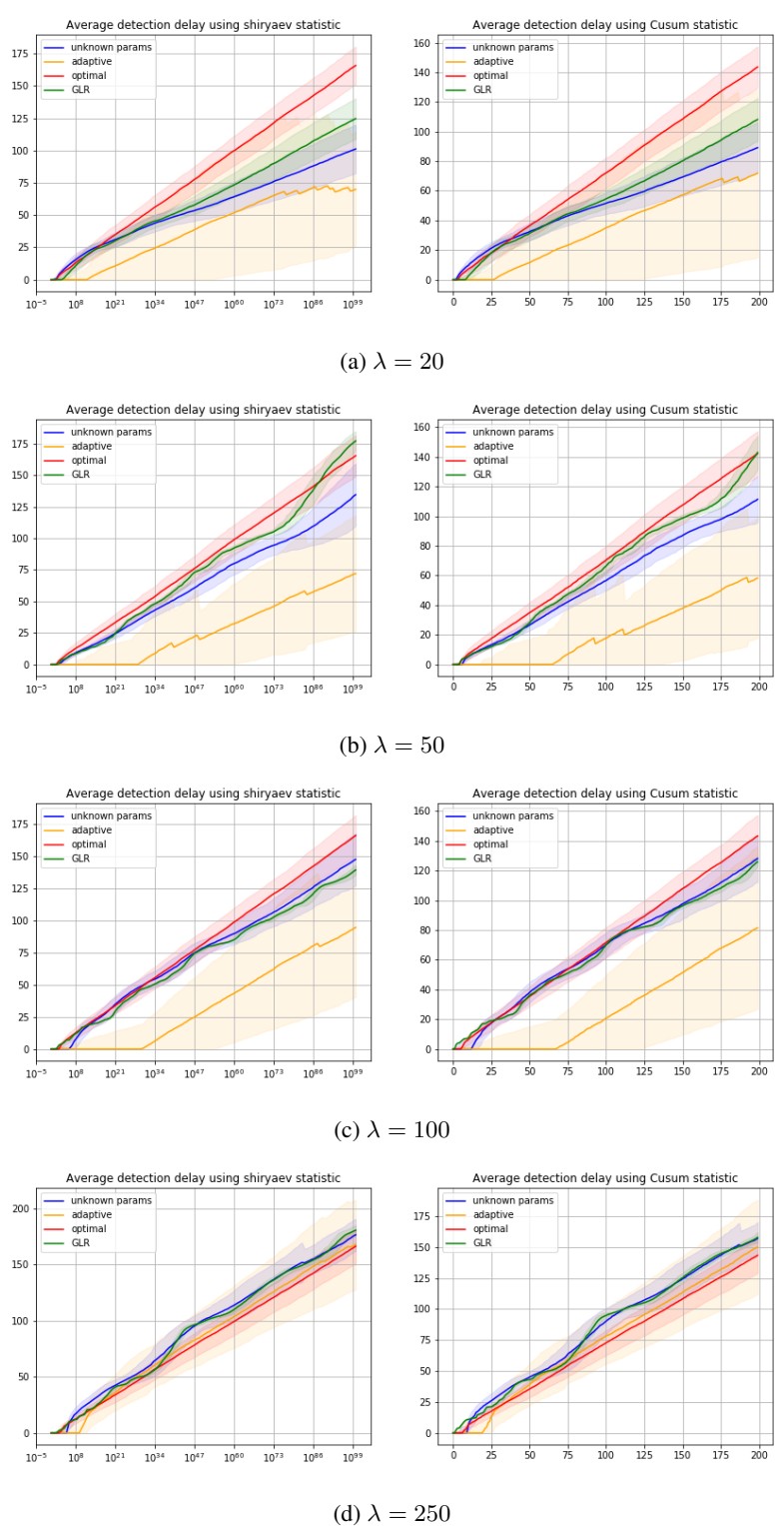

(a) $\lambda = 20$

(b) $\lambda = 50$

(c) $\lambda = 100$

(d) $\lambda = 250$

Figure 10: Performance analysis of the average detection delay as a function of the cutting threshold

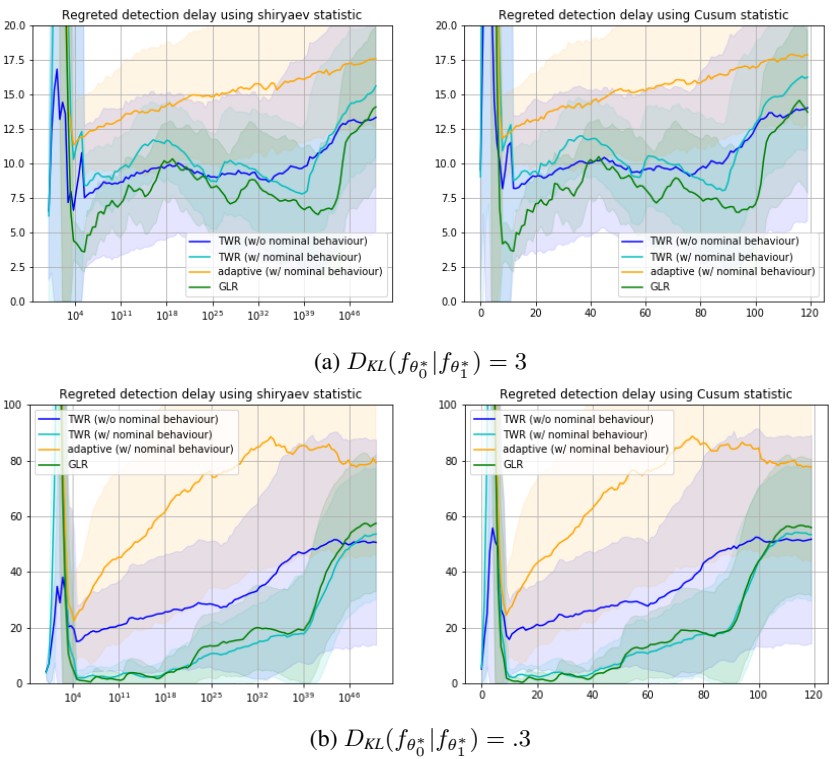

(a) $D_{KL}(f_{\theta_0^*}|f_{\theta_1^*}) = 3$

(b) $D_{KL}(f_{\theta_0^*}|f_{\theta_1^*}) = .3$

Figure 11: Performance analysis of the regretted detection delay given a nominal behaviour

PERFORMANCE GIVEN A NOMINAL BEHAVIOUR

In this section, we keep using the same hyper-parameters from the previous experiments. However we consider the setting where we have access to the pre-change parameter (which correspond to a nominal behaviour). Beside the **ADD** of the GLR and TWR statistics (computed with no prior knowledge of the pre-change parameter), we computed the **ADD** of the adaptive and TWR statistics using the prior knowledge of the true pre-change distribution.

The first notable thing is that the adaptive algorithm performances (yellow curve) did not improve by much compared to the case where the parameter were estimated using $10\%$ of the available observations. In fact TWR statistics outperforms it even with no prior knowledge of the nominal behaviour (dark blue curve). The second notable observation, is that given the pre-change parameter, TWR statistic performance (light blue) is within a constant of the GLR statistic. These observations confirm again that using the asymptotic behaviour of the Shriyaev detection delay is a very powerful approximation

