# OpenReview forum: "Quickest change detection for multi-task problems under unknown parameters"
_ICLR.cc/2021/Conference — Reject_

### Official Review · AnonReviewer4 · 2020-10-23
**Recommendation to Accept**

**Rating:** 7
**Confidence:** 4

**Review:**

This paper studies the quickest change detection for Markovian data, when both the parameters of pre- and post-change distributions are unknown. The main contribution is a scalable algorithm that sequentially estimates the unknown parameters and plug-in to classical detection schemes to get the stopping rule. A notable feature is that this is a joint estimation and detection framework. And the authors incorporate several tools, like SGD, annealing, penalization, into the detection task, which turns out to have good performance compared with existing benchmarks.

Overall, this paper is clearly-written and well-organized, and the numerical examples support the claims made in the paper.

Minor comments:
1. Usually in classical change-point detection literature, people assume the pre-change distribution is known since it can be estimated from historical (nominal) data, and the framework proposed in this paper can obviously be applied in such a setting as well. Therefore, I think it might be interesting to add one comparison in such setting (i.e., only post-change parameters is unknown and need to estimated). In such a case, the GLR and adaptive methods do not need to learn theta_0 offline and we can have a fair comparison of the performance of learning post-change parameters and also the detection delay.

2. In Appendix A.1, the introduction to SHIRYAEV Algorithm, it seems that there is a missing \rho in the denominator of the statistics S. The reason is that only under this \rho-scaled version of likelihood-ratio can the recursion in A.1 holds.

--------- After rebuttal ---------
Thanks to the authors for the response and updated paper. I keep my original score and recommend acceptance for this paper.

---

> ### Author Response · Authors · 2020-11-14
> **We thank the reviewer for his appreciation and thank them for their suggestion.**
>
> The proposed experiment is quite interesting as it benchmarks our algorithm's performances in the usual, simpler framework (where pre-change parameter are known). We even found out the following results:
> - TWR (without prior knowledge of the pre-change distribution) still beats the adaptive algorithm (with prior knowledge of the pre-change distribution)
> - TWR (with prior knowledge of the pre-change distribution) has a small constant delay with respect to the GLR approach!
>
> We will include this in the next update for the paper
>
>
> Considering the second minor remark, there is indeed a typo in the expression of $S_n^{\theta_0,\theta_1}$ that should read:
>     \begin{equation*}
>         S_n^{\theta_0,\theta_1} = \frac{1}{(1-\rho)^n} \sum_{k=1}^n (1-\rho)^{\textbf{k-1}} \prod_{t=k}^n L_t(\theta_0,\theta_1)
>     \end{equation*}
> Thanks for spotting it !

---

### Official Review · AnonReviewer2 · 2020-10-28
**A relevant topic but the paper falls short of acceptance level**

**Rating:** 7
**Confidence:** 4

**Review:**

The author(s) propose a quickest change detection technique under known parameter scenario. They use a Markovian dynamics to generate the pre and post change-point distributions and use a Shirayev test-statistic based on the asymptotic behavior of the optimal delay under know parameters. The proposed methodology is validated on synthetic data and a multitask reinforcement learning example. There are several issues which restricts the paper to reach an optimal level. These are highlighted below

-Heavy dependence on Tartakovsky and Veeravalli (2005) results with known parameters. The contribution in the current paper seems a bit incremental.

-The algorithm 1 looks promising however some of the hyper parameter choices such as c, $B_\alpha$, $\epsilon$ and $N_e$ are not clear.

-For a practitioner which one is better to choose- Shirayev test-statistic or Cusum?

-It was not quite clear to me why does one need both annealing and penalisation? I thought adjusting underestimation/overestimation of ${L_t}^*$ will be sufficient as behavior of pre-changepoint will complement the behavior post change-point.

-Does other choice of entropy based loss functions matter instead of KL divergence or KL divergence is the most natural choice here?

-

---

> ### Author Response · Authors · 2020-11-14
> **We understand the reviewer's concern, and hope the following answers provide a satisfying response.**
>
> **Heavy dependence on Tartakovsky and Veeravalli (2005) results with known parameters. The contribution in the current paper seems a bit incremental.**
>
> What's fundamentally different from previous contributions in the field is that we are evaluating the Shiryaev stopping time rather actually detecting the change point. To this extent, [Tartakovsky and Veeravalli (2005)] is irrelevant.
>
> It's true that the asymptotic behaviour of the Shirayev algorithm is a corner stone of our approach (as we segment the data accordingly), but we do not believe that this reduces the value of our contribution.
>
> In addition, simply using the asymptotic behaviour does not solve the QCD problem (cf Lemma 3) and we need to add safety brakes (annealing and penalisation) to have a fully working, state-of-the-art approximate algorithm.
> We also provide experimental results as well as theoretical results that ground our reasoning.
>
> **The algorithm 1 looks promising however some of the hyper parameter choices such as c, $\epsilon$, $B_\alpha$ and $N_e$ are not clear.**
>
> We discuss the hyper-parameter selection in the Appendix E-1 and  C-2 in order to respect the page count limit imposed by ICLR. Roughly speaking, they can be of three different types.
>
> - optimisations parameters (number of epochs $N_e$, gradient step, ...)
> - problem dependent parameters ($\epsilon$ and $c$) (their choice depends on the KL divergence of the pre and post change distributions as well as the mixing times of the associated Markov chains)
> - practitioner dependent parameters ($B_\alpha$) (describes how much we tolerate false alarms)
>
> **For a practitioner which one is better to choose- Shirayev test-statistic or Cusum?**
>
> It depends on how pessimistic the practitioner is.
> If you are pessimistic, you should optimise the \textbf{WADD} and thus use the CuSum statistic. If you are optimistic you should optimise the \textbf{ADD} and thus use the Shirayev statistic.
>
> Personally, we prefer the CuSum statistic as it is slightly more stable.
>
> **It was not quite clear to me why does one need both annealing and penalisation? I thought adjusting underestimation/overestimation will be sufficient as behavior of pre-changepoint will complement the behavior post change-point.**
>
> The stopping time defined in Equation (9) suffers from two problems.
>     1)  using only pre change parameters before the true change point $\lambda$  leads to  over-estimating  the statistic
>     2) using corrupted data after the Shiryaev change point %under known parameters $\nu_\alpha^S$
>
> Basically, penalising the likelihood ratio when $\theta_t^0$ and $\theta_t^1$ are similar solves the first issue, while the annealing solve the second one by taking into account the delay of detecting the Shiryaev change point under known parameters $\mathbb{E}[|\nu - \nu_\alpha^S|]$.
>
> As a consequence, the two principles need to be implemented together to yield the best possible performance. We provide in Appendix C an impact and ablation analysis of both procedures.
>
> **Does other choice of entropy based loss functions matter instead of KL divergence or KL divergence is the most natural choice here?**
>
> It's possible to consider other forms for the loss function, however as the KL divergence appears directly in the expression of the asymptotic delay under known parameters, we took it as the most natural choice here.

---

> > ### Comment · AnonReviewer2 · 2020-11-15
> > **Happy with the Author(s) response**
> >
> > Thanks very much for all the clarifications especially the difference in contribution in comparison to the Tartakovsky and Veeravalli (2005) paper. I still would appreciate a bit of discussion with there divergence measures. I understand the page limit constraint and also the fact that KL divergence appears naturally in the asymptotic results however some remark on divergence measures would be useful for readers!
> >
> > I changed my rating to "Accept".

---

> > > ### Author Response · Authors · 2020-11-17
> > > **We thank the reviewer for this support**
> > >
> > >
> > > We agree that discussing alternative divergence measures that are coherent with the QCD objective are indeed useful for the readers. We introduced a new subsection in the appendix (C-3 in the latest version), where we modify the loss function in order to optimise  $E_k[g(\frac{f_{\theta_1}}{f_{\theta_0}})]$ which is, under mild assumptions, similar to optimising an f-divergence measure where $f(x)=x.g(x)$.
> > >
> > > We also included empirical evaluation of the **ADD** using different f-divergences:
> > > - $g(x) = \log(x)$ (the KL divergence)
> > > - $g(x) = \sqrt{x}-1$
> > > - $g(x) = (x-1)*\log(x)$
> > >
> > > We provided the adaptive and the GLR average detection delay as a baseline.
> > > The experiments uncover yet again the great performances of our algorithm.

---

### Official Review · AnonReviewer3 · 2020-10-29
**Paper is marginally below acceptance threshold**

**Rating:** 7
**Confidence:** 2

**Review:**

*****  Paper's Summary  *****

This paper considers the quickest change detection (QCD) problem where pre-change and post-change distributions are unknown. For such problems, the authors proposed approximate algorithms in MIN-MAX and Bayesian settings. The algorithms run in O(1) and have near-optimal performances. The performance of proposed algorithms is verified using synthetic data and a reinforcement learning environment.


***** Paper's Strengths *****

The proposed algorithms are the approximate methods that have a near-optimal performance for QCD problems with unknown pre-change and post-change distributions.

The proposed algorithms are scalable and having low detection delays. Further, these algorithms work for a more general class of problems as they do not require restrictive conditions like IID samples, specific distributions, etc. on the problems.

The performance of proposed algorithms is better than existing algorithms.


***** Paper's Weaknesses *****

The proposed algorithms solve an optimization problem (depending on the setting) for minimizing the delay in change point detection. As no deep learning models (or even a variant) is used to solve the change point detection problem considered in the paper, this paper seems to be outside of the ICLR scope.

I find it very difficult to understand the paper in even 2-3 read. The authors need to improve overall writing quality so that it becomes easier to read and understand.


***** Comments *****

Some notations are not defined upfront e.g., Line 2 on Page 3: $S_n^{\theta_0, \theta_1}$ and $B_\alpha$.


*****  Questions for the Authors  *****

Please comments on how your paper fits the ICLR scope.


*****   Post Rebuttal  *****

Thank you for your clarifications! After reading the rebuttal and comments of other reviewers, I am increasing my score.

---

> ### Author Response · Authors · 2020-11-14
> **We thank the reviewer for his feed-back, and hope the following answers satisfy their concern.**
>
> **The proposed algorithms solve an optimisation problem (depending on the setting) for minimising the delay in change point detection. As no deep learning models (or even a variant) is used to solve the change point detection problem considered in the paper, this paper seems to be outside of the ICLR scope.**
>
> We respectfully disagree with the reviewer regarding this: the call for paper welcomes "submissions from all areas of machine learning and deep learning" (and even more precisely, solving the scalability issues of QCD problems falls in the "implementation issues" category).
>
> Change point problems have also been deemed, in the past, as a natural fit for ICLR:
>
>     - Pyramid Recurrent Neural Networks for Multi-Scale Change-Point Detection [ICLR 19']
>     - Kernel Change-point Detection with Auxiliary Deep Generative Models [ICLR 19']
>     - Bayesian Time Series Forecasting with Change Point and Anomaly Detection [ICLR 18']
>
> We hope that this convinces you that our work is relevant to the ICLR community and hope you would revise your score according as this relevancy issue seemed to be your main concern.
>
> **I find it very difficult to understand the paper in even 2-3 read. The authors need to improve overall writing quality so that it becomes easier to read and understand.**
>
> As the other reviewers didn't point out difficulties reading the paper (some complemented its quality actually), please point out unclear parts of the paper so that we can revisit them for a better readability.
>
> **Some notations are not defined upfront**
>
> We define formally $S_n^{\theta_0,\theta_1}$ and $B_\alpha$ in the Appendix A-1. However given the page limit in the main document we couldn't fit them upfront and we restricted ourselves with some insight about the nature of these quantities. We will point the reader in the next version to appendix A-1 for further details.

---

### Official Review · AnonReviewer1 · 2020-11-04
**Interesting idea, not very strong results**

**Rating:** 6
**Confidence:** 3

**Review:**

This paper studies the change point detection problem. The classical studies in change detection problems are based on the known prior and posterior parameters, i.e., knowing the distribution (parameters) before and after the change points. Recently, people are extending the results to the case where the prior parameter is known and the posterior parameter is unknown (anomaly detection) or with some sampling cost constraints (data-efficient change detection). However, this work proposes an algorithm that generalizes the CUSUM approach to the case where the parameters are unknown. The idea is very interesting and I believe the impact of the algorithm could be of significance given its potential in real-world applications. Besides, I have the following comments.

1) The paper title is for multi-rask problems. However, it seems to me that the proposed algorithm is very general for change detection problem. Except the one subsection in the experiments, I didn't see much connection to multi-task problems.

2) The theoretical results are not very strong. There is no Theorem one can claim for the performance of the proposed algorithm. As the algorithm is an approximation to some optimal approach, one may provide a result in the form of competitive ratio or convergence rate. However, Lemma 3 is only some asymptotic behavior of the loglikelihood.

3) How should one choose the hyperparameters like c and epsilon? Are the results in section 5 tuned by grid search and presented the best one?

---

> ### Author Response · Authors · 2020-11-14
> **We understand the reviewer's concern, and hope the following answers provide a satisfying response.**
>
> **The paper title is for multi-rask problems. However, it seems to me that the proposed algorithm is very general for change detection problem. Except the one subsection in the experiments, I didn't see much connection to multi-task problems.**
>
> We agree with the reviewer that our work is very general and can have a multitude of other possible applications. However, the QCD literature usually assume a nominal pre-change behaviour. Multi-task problems are a setting in which this assumption does not hold up.
>
> A crucial feature for multi-task RL is obviously the ability to detect change points in a tractable way. This is what brought us to this problem. Many other motivating examples are introduced in Appendix.
>
> However if you still find the title misleading, we propose to remove the 'multi-task' part from the title.
>
> **The theoretical results are not very strong. There is no Theorem one can claim for the performance of the proposed algorithm. As the algorithm is an approximation to some optimal approach, one may provide a result in the form of competitive ratio or convergence rate. However, Lemma 3 is only some asymptotic behavior of the loglikelihood.**
>
> We agree that our results do not provide strong non-asymptotic convergence guarantees. However, our theoretical results  (Lemma 1-3) justify that our algorithm efficiently approximates $L_t^*$. We believe that this in itself is an asymptotic performance guarantee:
>
> In fact, evaluating the performance of our algorithms can boil down to evaluating the detection delay of the Shiayev stopping time $ \mathbb{E} [| \nu - \nu_\alpha^S |] $. In our setting, this quantity is arguably proportional to the estimation error  $\mathbb{E}[|S_n^{\theta^0_t, \theta^1_t} - S_n^{\theta_0^*, \theta_1^*}|]$. which in turn is proportional to $\mathbb{E}[|L_t - L_t^*|]$.
>
> However deriving explicitly this relationship is not only cumbersome (integrating over all possible stopping time scenarios), but also require additional assumptions over the convergence rate of Equation 5.
>
> For these reasons, we chose to defend our algorithm with theoretically grounded insights and experimental results (over a wide variety of settings) rather than additional assumptions over this speed of convergence.
>
> **How should one choose the hyper-parameters like c and epsilon? Are the results in section 5 tuned by grid search and presented the best one?**
>
> We provide in Appendix E-1 some insight over the best practice to use when selecting these parameters. In a nutshell they depend on how much variability is expected in your data after the change (True KL divergence between pre- and post- change distributions) and on the stationarity of your distributions (Mixing time).
>
> The hyper-parameters in the presented results have been indeed tuned by grid-search.

---

### Decision · Program_Chairs · 2021-01-07
**Final Decision**

**Decision:**

Reject

**Comment:**

The paper treats a relevant and challenging problem in sequential learning scenarios -- how to detect distributional change over time when the pre- and post-change distributions are not known up to certainty. All reviewers more or less acknowledge that the paper presents a new approach towards solving this inference problem, where the high level idea is to approximately learn the pre- and post-change distribution parameters online using gradient descent and then apply well-known tests for change detection (e.g., the Shiryaev or CUSUM rules) with these assumed to be the pre- and post-change parameters.

However, beyond the concerns expressed by the reviewers, my finding after going through the manuscript myself is that the presentation of the paper's results leaves a lot to be desired in terms of clarity of exposition, comprehensiveness of performance benchmarking and comparison to existing approaches. Despite some of the reviwers' scores being revised upwards, the overall evaluation of the paper according to me is not adequate to merit acceptance, as per the concerns listed below.

1. There are two settings assumed in the paper (beginning of Sec. 2): (a) a completely Bayesian one, with the pre- and post-change distributional parameters drawn from a prior \cal{F} and the change time lambda drawn from a prior pi, and (b) a minmax one, where everything is the same as in (a) except that there is no prior over the change time lambda. However, it is not at all clear, in the algorithm design of the paper, where the prior \cal{F} over the distributions is used in computing (or approximating) conditional probabilities such as P[lambda | v_alpha = n].

2. There seem to be meaningless (or ill-defined) expressions in the paper's crucial portion motivating the algorithm design, such as P(X_t ~ f_{theta_0} | v_alpha = n), P(X_t ~ f_{theta_1} | v_alpha = n). It is hard to understand what the event "X_t ~ f_{theta_0}" even means -- I find it impossible to relate it to a sample path property. This leads me to question the validity of the technical development in the paper.

3. Another undefined term is "r-quickly" in eq. (4); I had to dig through the classical work of Lai, and Tartakovsky-Veeravalli to get a formal definition for this term. This is not to be expected of a paper that attempts to develop a new change point detection procedure from scratch, especially to an audience (ICLR) that may largely be unfamiliar with classicalt change detection theory.

4. There are several technical statements made without adequate formal proof, e.g., "Given the optimal stopping time \nu, it's possible to evaluate the posterior distribution of the change point P(lambda=t | v_alpha=n), which in turn is a good classifier of the pre and post change observation". What the precise meaning of the term "classifier" is, what its "goodness" is, and how exactly it is related to the posterior distribution of lambda given the value of v_alpha, is formally not spelt out for a paper that largely uses formal probability language to develop its main results.

5. While I understand that the final algorithm to detect the change involves several approximations and heuristics along the way, which may very well be intuitively appealing, I do not understand (even after repeated passes over the submission) several key aspects -- a concern also expressed by Reviewer 3. Why is it reasonable to assume that the conditional distribution of the change time lambda given the algorithm's stop time v_alpha would be logistic, and with the specific parameters mu and s given in the section "Distribution Approximation"? Moreover, it is hard to discern from the crucial Section 3.2 why the functions f_0^n, f_1^n should be useful in practice as proxies to the actual expected log likelihood ratios under the true parameters -- despite Lemma 2 showing that they converge to the true expectations (again, the sense in which this convergence occurs is omitted leading to imprecision in the statement), the rates as a function of n, t_2 may be slow. I agree in this regard with the same concern voiced by Reviewer 1, and do not see a satisfactory explanation to it in the paper's discussion.

6. Comparison to literature. Contrary to the general picture painted in the paper about the lack of sufficient investigation of the "unknown pre and post change parameter" case, there does seem to be a rigorous body of work existing in this line that is not discussed in the manuscript. For instance, "SEQUENTIAL CHANGE-POINT DETECTION WHEN THE PRE- AND POST-CHANGE PARAMETERS ARE UNKNOWN", Lai and Xing, 2009, and "A BAYESIAN APPROACH TO SEQUENTIAL SURVEILLANCE IN EXPONENTIAL FAMILIES", Lai-Liu-Xing, 2009, are both works that address this very setting and in a comprehensive manner with theoretical guarantees. What the current manuscript does, in the context of both these works, is highly unclear. Is it trying to suggest an approximate way of computing the natural posterior distribution of the change time lambda given all data up to now, using the proxy P(lambda | v_alpha = n), or using a completely different approach altogether, is not adequately discussed at all, which makes the motivating arguments for the algorithm vague.

7. Finally, but in no lesser measure, the Experimental Results section features a rather narrow set of (two) scenarios for which it presents numerics. For a paper that claims to demonstrate "experimental results (over a wide variety of settings)" [from the author response], this is quite telling as it renders the argument in favor of the paper's approach quite weak. Here again, for the first (synthetic) setting, I do not understand the relevance of the neural network adopted to fix the parameters of a Gaussian distribution. Moreover, the reported distributions of the "regretted detection delay" seem to be quite wide for all the approaches compared (unknown params, adaptive, GLR), precluding a reasonable comparison of their performance. The author(s) would do well to expand the scope of both synthetic and non-synthetic experiments to show the validity of their approach, and in each case carry out many more independent trials than just 500 for more accurate benchmarks.

I do note that more experimental results have been reported in the appendix, but I would presume that they have more value being in the main body after the algorithm design is explained in a more succinct and clearer manner. This can only come about by a significant rewriting and reorganizing of the paper, which I am confident the author(s) can carry out in order to make this into a much stronger submission. I wish the author(s) good luck on this, and hope to see the strengths of this new approach brought out in a more impactful manner in the next revision.